# IMPLICIT 4D GAUSSIAN SPLATTING FOR FAST MOTION WITH LARGE INTER-FRAME DISPLACEMENTS

**Seung-gyeom Kim**  **Areum Kim**  **Yongjae Yoo**  **Sukmin Yun**[*]
Department of Applied Artificial Intelligence, Hanyang University
{skkim3533, dkfma0817, yongjaeyoo, sukminyun}@hanyang.ac.kr

## ABSTRACT

Recent 4D Gaussian Splatting (4DGS) methods often fail under fast motion with large inter-frame displacements, where Gaussian attributes are poorly learned during training, and fast-moving objects are often lost from the reconstruction. In this work, we introduce Spatiotemporal Position Implicit Network for 4DGS, coined SPIN-4DGS, which learns Gaussian attributes from explicitly collected spatiotemporal positions rather than modeling temporal displacements, thereby enabling more faithful splatting under fast motions with large inter-frame displacements. To avoid the heavy memory overhead of explicitly optimizing attributes across all spatiotemporal positions, we instead predict them with a lightweight feed-forward network trained under a rasterization-based reconstruction loss. Consequently, SPIN-4DGS learns shared representations across Gaussians, effectively capturing spatiotemporal consistency and enabling stable high-quality Gaussian splatting even under challenging motions. Across extensive experiments, SPIN-4DGS consistently achieves higher fidelity under large displacements, with clear improvements in PSNR and SSIM on challenging sports scenes from the CMU Panoptic dataset. For example, SPIN-4DGS notably outperforms the strongest baseline, D3DGS, by achieving +1.83 higher PSNR on the Basketball scene.

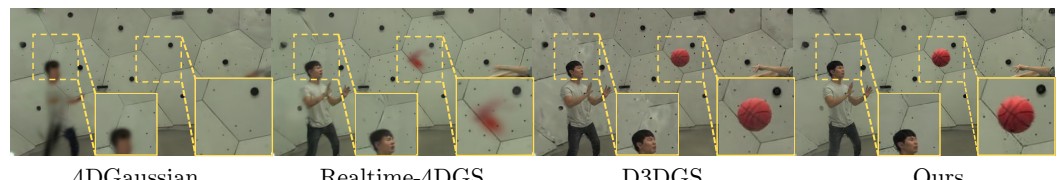

|            4DGaussian | Realtime-4DGS | D3DGS | Ours |

Figure 1: **Faithful reconstruction of fast motion with large inter-frame displacements.** Existing 4DGS approaches often produce blurred or incomplete reconstructions of fast-moving objects. In contrast, ours successfully reconstructs clear and accurate details, such as the basketball in the scene.

## 1 INTRODUCTION

Rendering fast motions with large inter-frame displacements remains challenging for dynamic scene reconstruction, despite its importance for a wide range of real-world applications. Recent advances in 4D Gaussian Splatting (4DGS) have shown remarkable efficiency and visual quality, making it a promising framework for dynamic scene reconstruction. In particular, existing 4DGS methods (Yang et al., 2024a; Duan et al., 2024; Wu et al., 2024; Xu et al., 2024) achieve strong results on dynamic scenes with small displacements (*e.g.*, Neu3D (Li et al., 2022b)) across video frames.

However, as motions become faster and inter-frame shifts grow larger (*e.g.*, Panoptic Sports (Joo et al., 2015)), existing 4DGS methods, including explicit 4D parametrization (Yang et al., 2024a; Duan et al., 2024) and deformable approaches (Wu et al., 2024; Xu et al., 2024; Kwak et al., 2025), often fail to capture the rapid dynamics, producing blurred or even vanished objects. To be specific, in deformable approaches, Gaussians are defined in a static canonical space and transformed over time through learned deformations. However, they often fail to assign initial Gaussians for fast-moving objects in the canonical space, causing those objects to remain unseen during deformation

---

*Corresponding author. Also affiliated with Hanyang University ERICA.

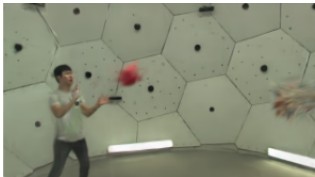 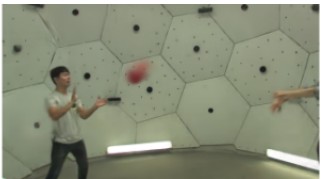 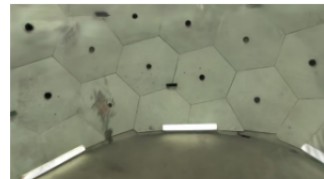

(a) **Left:** 15K training iteration, **Right:** 30K training iteration.  (b) Canonical space

Figure 2: **Failure modes on fast motions with large inter-frame displacements.** We visualize failure modes of existing frameworks; (2a) explicit parameterization and (2b) deformable methods. Figure (2a) shows drastic degradation on training iterations (*i.e.*, $15K \rightarrow 30K$), and (2b) shows the canonical space of deformable initialization fails to assign Gaussians for fast motions.

training. On the other hand, although explicit parameterization roughly tracks Gaussian positions that correspond to fast motions at the early stage of training, their attributes, including color, opacity, scale, and rotation, rapidly collapse in later training, leading to drastic degradation. As a result, both approaches show failure modes with blurred or even vanished motions, as shown in Figures 1 and 2.

To this end, we focus on addressing the failure in learning Gaussian attributes for fast-moving objects with large displacements. Although positions remain sufficiently accurate to capture motion, other attributes collapse more easily. This degradation arises because reconstruction loss is dominated by background Gaussians; fast-moving Gaussians at new positions incur higher reconstruction errors, while static backgrounds are easier to fit. As a result, both deformable and explicit 4DGS approaches tend to bias learning toward background fitting, eventually leading dynamic objects to disappear. In addition, such large displacements can cause cross-frame interference during frame-by-frame rasterization. Although a Realtime-4DGS (Yang et al., 2024a) can be sliced differently at each time, its parameters are shared, so optimizing for one frame makes other slices suboptimal unless we separate Gaussians by explicit spatiotemporal positions $(x, y, z, t)$. This observation motivates us to leverage the explicit spatiotemporal positions of fast-moving Gaussians as inputs for generating their attributes, thereby achieving more faithful splatting under large displacements.

In this paper, we introduce SPIN-4DGS (**S**patiotemporal **P**osition **I**mplicit **N**etwork for 4DGS), a lightweight yet effective framework designed to handle fast motions with large inter-frame displacements. To be specific, we first estimate high-quality spatiotemporal positions $(x, y, z, t)$ of Gaussians that can serve as the inputs for later attribute prediction, initially gathering them across the entire scene before refining them in a frame-wise manner. Then, we leverage these positions to predict Gaussian attributes through a lightweight feed-forward network, avoiding the heavy memory overhead of explicitly optimizing attributes over all spatiotemporal positions. This design learns a shared implicit representation across all Gaussians and decodes attributes directly from positions under rasterization loss. As a result, learned attributes remain consistent across positions, and dynamic objects can stay stable even under large displacements. Meanwhile, attributes are stored implicitly in network parameters, rather than explicitly for each Gaussian, which significantly improves memory efficiency on a large number of spatiotemporal positions.

To validate the effectiveness of the proposed SPIN-4DGS, we perform experiments on various sports scenes from the CMU Panoptic Sports dataset, where human motions are rapid and small objects move across large inter-frame displacements. Across six sports scenes, SPIN-4DGS achieves the best performances, significantly outperforming all baselines. For example, SPIN-4DGS achieves a +1.83 PSNR dB improvement compared to the strongest baseline, D3DGS (Luiten et al., 2024), with a higher SSIM of 0.92 on the Basketball scene. Specifically, qualitative results in Figure 4 and Table 1 demonstrate that prior methods often blur Gaussians or fail to capture fast-moving objects (*e.g.*, Basketball), whereas SPIN-4DGS preserves them with sharp and stable reconstructions. Interestingly, we further observe that SPIN-4DGS significantly improves performance when reusing pre-trained Gaussian positions from strong baselines, such as D3DGS and Realtime-4DGS.

Overall, our work introduces SPIN-4DGS, the first framework that learns Gaussian attributes directly from jointly optimized frame-wise positions, enabling stable and high-quality 4DGS under large inter-frame displacements. This 4D reparameterization allows ours to mitigate temporal failure modes under such fast motion while still capturing rich temporal appearance dynamics. We believe SPIN-4DGS addresses a fundamental challenge of 4DGS by enabling stable learning in various real-world scenarios, thereby opening a new direction for advancing dynamic scene representation.

## 2 RELATED WORK

**Learning temporal dynamics for 3D Gaussian.** Recent advances have shown significant progress in extending 3D Gaussian representations into the 4D domain by learning temporal dynamics. Early attempts (Luiten et al., 2024; Javed et al., 2024) incrementally propagated 3D Gaussians from the first frame, demonstrating the potential of 4D Gaussian splatting, but suffering from sequential inefficiency and overfitting due to numerous per-frame iterations. However, their reliance on external supervision, such as segmentation masks, substantially increases training costs and computational overhead. Meanwhile, inspired by NeRF approaches (Cao & Johnson, 2023; Fridovich-Keil et al., 2023; Park et al., 2021), deformable frameworks (Wu et al., 2024; Yang et al., 2024b; Lu et al., 2024; Zhu et al., 2024; Lin et al., 2024; Xu et al., 2024) instead construct a canonical space to initialize Gaussians and then learn temporal displacements in rotation, scale, and position. Despite their robustness on motions with small inter-frame displacements, these methods still underperform on rapid motions with large displacements, as Gaussians that represent fast motion are often omitted during canonicalization and thus remain unseen in the subsequent deformation process. In contrast, we address these challenges without external supervision, showing that high-fidelity capture of fast motion can be achieved solely from raw video inputs.

**Explicit 4D Gaussian Parameterizations.** Another promising direction is to focus on directly parameterizing Gaussians in 4D from scratch (Yang et al., 2024a; Lee et al., 2024; Duan et al., 2024), rather than optimizing each frame separately in the 3D domain. Such explicit 4D modeling unifies space and time into a continuous field and encodes dynamics as 4D Gaussian splats, which can be temporally sliced for rendering, in contrast to deformable-based approaches. For instance, Realtime-4DGS (Yang et al., 2024a) performs temporal slicing of 4D Gaussians at each timestamp to obtain dynamic 3D Gaussians, which are then projected to the image plane. While explicit 4D parameterizations achieve higher frame rates (FPS) and improved rendering quality, they require longer training times and larger storage requirements. Moreover, under fast motion with large displacements, we observed that corresponding Gaussian attributes gradually blur due to cross-frame interference. A single 4D Gaussian must simultaneously explain all timestamps; however, rasterization is performed frame by frame, so optimization for one frame inevitably affects the others. This inherent conflict in the rasterization process makes it challenging to maintain temporal consistency in dynamic regions. In contrast, our method avoids such cross-frame degradation by directly decoding Gaussians at explicit spatiotemporal positions in a feed-forward manner. This design explicitly separates large-displacement Gaussians at each timestamp, which prevents cross-frame degradation and leads to both stable training and consistent rendering quality.

## 3 METHOD

In this section, we present the Spatiotemporal Position Implicit Network for 4DGS (SPIN-4DGS), a framework for reconstructing dynamic scenes under motions with large inter-frame displacements. We first review 3D Gaussian Splatting as preliminaries in section 3.1. Then, section 3.2 describes how we obtain explicit spatiotemporal Gaussian positions, and section 3.3 details how their attributes are predicted via a feed-forward implicit network. The overall framework is illustrated in Figure 3.

### 3.1 PRELIMINARY: 3D GAUSSIAN SPLATTING

3D Gaussian Splatting (3DGS; Kerbl et al. (2023)) provides a differentiable volume rendering representation. It introduces anisotropic Gaussians where the parameters (*i.e.*, position, scale, rotation, color, and opacity) of each Gaussian are defined and optimized for tile-based rendering. Structure-from-Motion (SfM; Schonberger & Frahm (2016)) techniques estimate the initial positions and colors of Gaussians from input images. For a given arbitrary point $\mathbf{x} \in \mathbb{R}^3$ in the 3D scene, Gaussian is defined as follows:

$$G^{3D}(\mathbf{x}) = \exp\left(-\frac{1}{2}(\mathbf{x} - \boldsymbol{\mu})^\top \boldsymbol{\Sigma}_{3D}^{-1}(\mathbf{x} - \boldsymbol{\mu})\right), \tag{1}$$

where $G^{3D}(\mathbf{x})$ denotes the value of the Gaussian at arbitrary point $\mathbf{x}$. The parameters of each Gaussian include the position $\boldsymbol{\mu} \in \mathbb{R}^3$, opacity $o \in \mathbb{R}$, color $\mathbf{c} \in \mathbb{R}^3$ represented using spherical harmonics coefficients, and the covariance matrix $\boldsymbol{\Sigma} \in \mathbb{R}^{3 \times 3}$, which is defined in terms of a diagonal

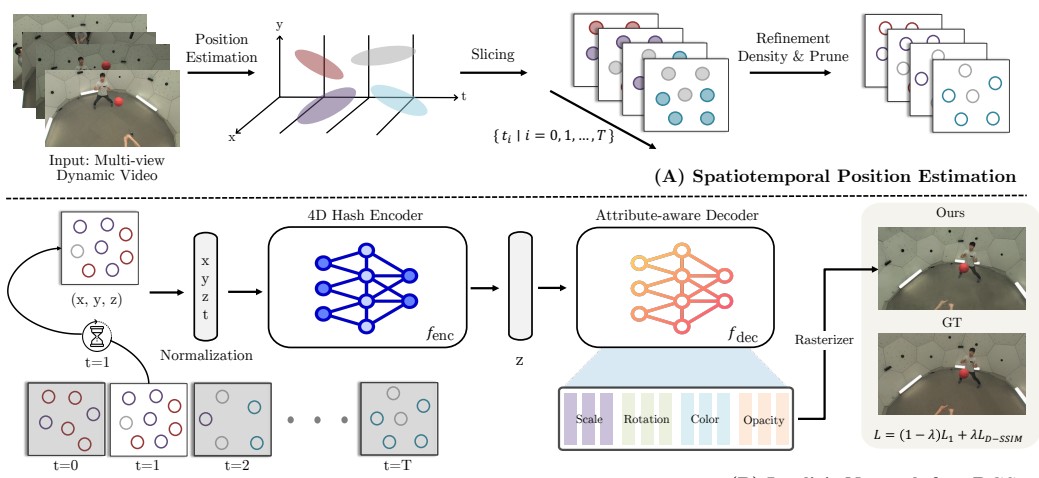

**Figure 3: Illustration of the overall framework.** SPIN-4DGS consists of two stages of (a) Spatiotemporal Position Estimation and (b) Implicit Network for 4DGS. Specifically, (a) we slice Gaussians along the temporal axis to obtain spatiotemporal position sets and refine them with rasterization loss. Then, (b) the refined positions are normalized and passed through a 4D hash encoder and multibranch decoders to predict Gaussian attributes (scale, rotation, color, and opacity).

scale matrix $\mathbf{S} = \mathrm{diag}(s_1, s_2, s_3)$ and a rotation matrix $\mathbf{R} \in \mathrm{SO}(3)$, obtained from quaternions. To ensure that the covariance matrix remains $\mathbf{\Sigma}$ positive semi-definite, it is computed as follows:

$$\mathbf{\Sigma}_{3D} = \mathbf{R}\mathbf{S}\mathbf{S}^\top\mathbf{R}^\top. \tag{2}$$

To render via rasterization, the 3D Gaussians are first projected onto the 2D image plane. This is done by applying the viewing transformation $\mathbf{W}$ and the Jacobian matrix (Zwicker et al., 2001) $\mathbf{J}$ to compute the 2D covariance matrix $\mathbf{\Sigma}_{2D}$ as follows:

$$\mathbf{\Sigma}_{2D} = \mathbf{J}\,\mathbf{W}\,\mathbf{\Sigma}_{3D}\,\mathbf{W}^\top\mathbf{J}^\top \tag{3}$$

During the rendering process, pixel values are computed via alpha blending. The alpha value (*i.e.*, opacity) of each Gaussian is obtained by projecting the 3D Gaussian $G_i$ into 2D as $G_i^{2D}$. Specifically, for each pixel, the alpha value $\alpha_i'$ and the resulting color $\mathbf{C}$ are computed as

$$\alpha_i' = o_i\, G_i^{2D}(\mathbf{x}), \quad \mathbf{C}(\mathbf{x}) = \sum_{i=1}^{N} c_i\, \alpha_i' \prod_{j=1}^{i-1}(1 - \alpha_j'), \tag{4}$$

where $o_i$ is the intrinsic opacity of the $i$-th Gaussian, $c_i$ its color, and $N$ the total number of Gaussians contributing to the pixel.

## 3.2 Spatiotemporal Gaussian Positions

In this section, our goal is to construct a Gaussian point set that fully spans the trajectory of dynamic objects, ensuring sufficient coverage for faithful reconstruction. Prior methods perform reasonably well in regions where Gaussians are densely clustered, but under large inter-frame displacements, they often fail to maintain enough points around fast-moving objects. Even when positions are sufficiently available, attributes remain challenging. To be specific, when Gaussians spanning the entire trajectory are optimized jointly, frame-specific supervision signals interfere with one another. Because rasterization is performed frame by frame, updates that make a Gaussian optimal for one timestamp can render it suboptimal for others. This cross-frame interference can potentially weaken the learning signal and degrade the quality of the reconstruction.

To overcome these issues, we construct Gaussian sets independently at each time step, explicitly separating them by spatiotemporal positions. This formulation avoids interference across frames and enables more reliable attribute learning for fast-moving objects under large displacements.

$$u_t = f_\theta(\mathbf{x}, \mathbf{y}, \mathbf{z}, \mathbf{t}), \qquad \mathbf{c}_t = g_\phi(\mathbf{x}, \mathbf{y}, \mathbf{z}, \mathbf{t}) \in \{0, \dots, 255\}^3, \qquad t \in \{0, \dots, T\}. \tag{5}$$

Here, $f_\theta$ and $g_\phi$ are explicit functions of the temporal axis predicting the position and color corresponding to each frame $\mathbf{t}$. For example, the explicit approaches (Duan et al., 2024; Yang et al., 2024a) construct points by performing time slicing along the temporal axis, whereas a deformable approach (Wu et al., 2024; Xu et al., 2024; Bae et al., 2024) network can learn to construct them as time-varying structures. For SPIN-4DGS, we estimate spatiotemporal positions by an explicit method (Yang et al., 2024a) as a default. Lastly, we further refine the estimated position by utilizing rasterization loss with corresponding colors at every frame to densify salient points and prune unnecessary ones.

$$u_t \;\leftarrow\; \text{Refine}\big(u_t, \mathbf{c}_t; t\big), \qquad t = 0, \ldots, T. \tag{6}$$

After refinement, the Gaussian points at each timestamp $t$ are fixed as explicit spatiotemporal positions and directly fed into our implicit network to learn the corresponding attributes.

### 3.3 IMPLICIT NETWORK FOR 4D GAUSSIAN SPLATTING

In this section, we describe an implicit network that is trained in an end-to-end manner to directly predict 4D Gaussian parameters from learnable spatiotemporal Gaussian positions initialized from the collected ones. In contrast to prior approaches that require pre-defined or pre-optimized Gaussian attributes (*e.g.*, scale, rotation, opacity, color) to model temporal dynamics, our network is trained from scratch using only the collected spatiotemporal positions $(\mu, t) \in \mathbb{R}^4$ as input. Given $(\mu, t)$, the network predicts all 4D Gaussian parameters, including opacity $o \in \mathbb{R}^1$, spherical harmonics coefficients $sh \in \mathbb{R}^{48}$, scale $\mathbf{s} \in \mathbb{R}^3$, and rotation represented as a unit quaternion $\mathbf{r} \in \mathbb{R}^4$.

**Input position normalization.** At each frame $t$, the spatiotemporal Gaussian positions serve as inputs to a feed-forward network for learning implicit representations of other Gaussian parameters. Since the raw 3D Gaussian positions $\mathbf{u} \in \mathbb{R}^3$ on the scene are unbounded, we apply a normalization step to stabilize learning and preserve representational capacity. Specifically, following the scene contraction strategy of Mip-NeRF (Barron et al., 2021), we first compress the coordinates into a finite ball and then map them to the normalized range $[0, 1]^3$, as defined in equation 7.

$$\text{contract}(\boldsymbol{\mu}) = \begin{cases} \boldsymbol{\mu}, & \|\boldsymbol{\mu}\| \leq 1, \\ \left(2 - \dfrac{1}{\|\boldsymbol{\mu}\|}\right) \dfrac{\boldsymbol{\mu}}{\|\boldsymbol{\mu}\|}, & \|\boldsymbol{\mu}\| > 1, \end{cases} \qquad \bar{\boldsymbol{\mu}} = \tfrac{1}{4}\text{contract}(\boldsymbol{\mu}) + \tfrac{1}{2} \in [0, 1]^3. \tag{7}$$

To keep spatial and temporal scales comparable, we normalize time to $[0, 1]$ to match the scale of the spatial embedding. Concretely, we use the current timestamp divided by the total duration:

$$t_{\text{norm}} \;=\; \frac{t - t_{\min}}{t_{\max} - t_{\min}} \;\in\; [0, 1] \quad \text{and} \quad \tilde{\mathbf{x}} \;=\; \big[\, \bar{\boldsymbol{\mu}}^\top, \, t_{\text{norm}} \,\big]^\top \;\in\; [0, 1]^4. \tag{8}$$

Our network adopts an encoder-decoder architecture, where the encoder $\text{f}_{enc}$ maps the normalized input $\tilde{\mathbf{x}}$ to a latent embedding $\text{f}_{enc}(\tilde{\mathbf{x}})$ which the decoder takes as input. We note that the position $\mathbf{u}$ is also parameterized and jointly optimized with the network, similar to existing frameworks (Wu et al., 2024; Xu et al., 2024).

**Encoder architecture for shared latent representation.** We directly extend the widely used 3D Instant-NGP (Müller et al., 2022) multi-hash grid to 4D by appending the temporal axis, producing a compact latent vector for each input. Given the normalized input $\tilde{\mathbf{x}} = [\,\bar{\boldsymbol{\mu}}^\top, t_{\text{norm}}\,]^\top$, we map it into a unified 4D embedding via a single spatio-temporal encoder. In contrast to low-rank decompositions (*e.g.*, planar), which may reduce computation but degrade expressiveness as the number of Gaussians grows and incur extra cost from per-level hash management, we avoid such factorizations and employ the 4D hash encoder (Chen et al., 2025) directly as follows:

$$z = \text{f}_{enc}(\tilde{\mathbf{x}}) \in \mathbb{R}^{LF} \tag{9}$$

where $L$ denotes the number of hash levels and $F$ the number of channels per level. We concatenate features across all levels to form $z \in \mathbb{R}^{LF}$, which we use as the latent representation.

**Attribute-aware decoder for Gaussian attribute prediction.** Given a latent vector $\mathbf{z}$, we use a multi-branch decoder to produce Gaussian parameters, scale, rotation, spherical harmonics coefficients, and opacity. Each attribute is predicted by a separate head (three-layer MLP) taking the

shared encoder output $\mathbf{z} \in \mathbb{R}^{LF}$ as input. All decoder heads use GELU (Hendrycks & Gimpel, 2016) activations instead of ReLU (Agarap, 2018).

$$(\hat{\mathbf{s}}, \hat{\mathbf{r}}, \hat{\mathbf{sh}}, \hat{\mathbf{o}}) = \big( f_{\text{scale}}(\mathbf{z}), \ f_{\text{rot}}(\mathbf{z}), \ f_{\text{sh}}(\mathbf{z}), \ f_{\text{opacity}}(\mathbf{z}) \big). \quad (10)$$

We convert the raw outputs into valid parameter ranges using attribute-specific activations and post-processing:

$$(\mathbf{s}, \mathbf{r}, \mathbf{sh}, \mathbf{o}) = \big( \exp(\hat{\mathbf{s}}), \ \tfrac{\hat{\mathbf{r}}}{\|\hat{\mathbf{r}}\|_2}, \ \hat{\mathbf{sh}}, \ \sigma(\hat{\mathbf{o}}) \big). \quad (11)$$

We also follow the Gaussian post-processing (Kerbl et al., 2023) pipeline, with a few additional steps for stable training.

**Scale decoder.** To prevent gradient explosion from exponential growth, we clip the pre-scale in the backward pass: $\hat{s} \leftarrow \text{clip}(\hat{s}, \max = 20)$. We also initialize the final-layer bias to $-5$ to start from a small scale, *i.e.*, $\exp(-5) \approx 0.0067$, and keep it trainable.

**Rotation decoder.** To bias the predicted quaternion toward the identity at the start of training, we set the final-layer bias to $(1, 0, 0, 0)$, *i.e.*, the first element to 1.0 and the remaining elements to 0, so that the initial output satisfies $\hat{\mathbf{r}} \approx (1, 0, 0, 0)$. This initialization lets the network learn rotations progressively while preserving a stable initial structure.

**Opacity decoder.** To stabilize early training and encourage a near-transparent start, we set the final-layer bias to $\text{logit}(0.1) \approx -2.197$ (trainable), initializing $\hat{o}$ at $\approx 0.1$. The network then learns to increase opacity only where needed, focusing on informative points.

**Color decoder.** The decoder directly regresses color coefficients from encoder embeddings, with no special initialization, to capture the high-dimensional color required for SH-based rendering.

**Loss objectives.** Finally, frame images are rendered via rasterization, and we optimize the model using the standard 3DGS reconstruction loss as follows:

$$\mathcal{L} = (1 - \lambda) \, \mathcal{L}_1 + \lambda \, \mathcal{L}_{\text{D-SSIM}}. \quad (12)$$

where $\lambda$ is a hyperparameter; we set $\lambda = 0.2$, following prior works (Kerbl et al., 2023).

## 4 EXPERIMENTS

In this section, we demonstrate the effectiveness of the proposed method, SPIN-4DGS. Specifically, we choose to employ the recent explicit parameterization method for Realtime-4DGS (Yang et al., 2024a), which is publicly available, only in the early training stage to estimate spatiotemporal Gaussian positions. These positions are then used in our framework to learn new Gaussian attributes. We then evaluate its ability to capture fast motion on various sports scene benchmarks from the CMU Panoptic Sports dataset (Joo et al., 2015), comparing it with existing 4DGS baselines.

**Implementation details.** The network uses a hidden dimension of 64, and the encoder is configured with $L = 16$ levels and $F = 4$ features per level; the hash map size $2^{21}$. Parameter groups in both the encoder and the decoder utilize separate learning rates, as Gaussian attributes (*e.g.*, scale and rotation) are sensitive, and a uniform learning rate often leads to unstable optimization. The encoder learning rate is initialized to $8 \times 10^{-3}$; decoder learning rates are $1 \times 10^{-3}$ for color, $3 \times 10^{-4}$ for scale, $3 \times 10^{-5}$ for rotation, and $8 \times 10^{-4}$ for opacity. Position parameters use the same learning rate as 3DGS. We use Adam (Kingma & Ba, 2015) with a linear warm-up and cosine decay schedule, under which every parameter's learning rate decays to $1 \times 10^{-7}$. Experiments were conducted on an RTX 4090 GPU (24 GB), and the implementation utilized PyTorch (Paszke et al., 2019) 2.1 with CUDA 11.8. By default, we train for 40K iterations with a batch size of 3. We perform quantitative evaluations using PSNR (Peak Signal-to-Noise Ratio), SSIM (Wang et al. (2004); Structural Similarity Index), and LPIPS (Zhang et al. (2018); Learned Perceptual Image Patch Similarity), and additionally report frames per second (FPS) to assess rendering speed.

**Datasets.** We employ the CMU Panoptic Sports dataset (Joo et al., 2015) to validate scenarios of fast motions with large inter-frame displacements in our experiments. Specifically, the Panoptic Sports dataset is a challenging benchmark containing six sports scenes: juggle, basketball, boxes, football, softball, and tennis. Each scene is recorded at 30 FPS for 5 seconds (150 frames per scene).

Table 1: **Comparisons on dynamic sports scenes in the CMU Panoptic Sports dataset.** We evaluate ours with existing 4DGS baselines on benchmarks containing fast motions with large inter-frame displacements. We report PSNR and SSIM for six sports scene sequences across all baselines.

| Method | 4DGS Category | Basketball | | Boxes | | Football | | Juggle | | Softball | | Tennis | | Avg. | | | |
|---|---|---|---|---|---|---|---|---|---|---|---|---|---|---|---|---|---|
| | | PSNR↑ | SSIM↑ | PSNR↑ | SSIM↑ | PSNR↑ | SSIM↑ | PSNR↑ | SSIM↑ | PSNR↑ | SSIM↑ | PSNR↑ | SSIM↑ | PSNR↑ | SSIM↑ | FPS↑ | Storage↓ |
| Grid4D (Xu et al., 2024) | Deformable | 25.82 | 0.89 | 26.81 | 0.91 | 27.61 | 0.91 | 28.09 | 0.92 | 26.91 | 0.91 | 27.10 | 0.91 | 27.06 | 0.91 | 146 | 333 |
| MoDec-GS (Kwak et al., 2025) | Deformable | 27.42 | 0.90 | 26.17 | 0.92 | 27.09 | 0.92 | 28.03 | 0.93 | 27.31 | 0.92 | 27.36 | 0.92 | 27.23 | 0.92 | 62 | **34** |
| 4DGaussian (Wu et al., 2024) | Deformable | 27.28 | 0.90 | 27.32 | 0.91 | 28.71 | 0.91 | 26.94 | 0.91 | 27.24 | 0.91 | 27.66 | 0.91 | 27.53 | 0.91 | 40 | 62 |
| TC3DGS (Javed et al., 2024) | External supervision | 27.92 | 0.89 | 28.28 | 0.89 | 28.00 | 0.89 | 29.15 | 0.90 | 27.96 | 0.89 | 25.97 | 0.89 | 27.88 | 0.89 | **890** | 49 |
| D3DGS (Luiten et al., 2024) | External supervision | 28.22 | 0.91 | 29.46 | 0.91 | 28.49 | 0.91 | 29.48 | 0.91 | 28.43 | 0.91 | 28.11 | 0.91 | 28.70 | 0.91 | 760 | 1994 |
| 4D-Rotor-Gaussians (Duan et al., 2024) | Explicit | 27.76 | 0.90 | 26.94 | 0.89 | 26.54 | 0.92 | 27.62 | 0.92 | 27.03 | 0.92 | 27.09 | 0.91 | 27.16 | 0.91 | 100 | 94 |
| Realtime-4DGS (Yang et al., 2024a) | Explicit | 27.89 | **0.92** | 28.17 | **0.93** | 28.35 | **0.93** | 28.68 | **0.93** | 28.67 | **0.93** | 28.50 | **0.93** | 28.38 | **0.93** | 197 | 1293 |
| Ours | | **30.05** | **0.92** | **29.91** | **0.93** | **29.99** | **0.93** | **30.31** | **0.93** | **30.24** | **0.93** | **30.14** | **0.93** | **30.11** | **0.93** | 104 | 1261 |

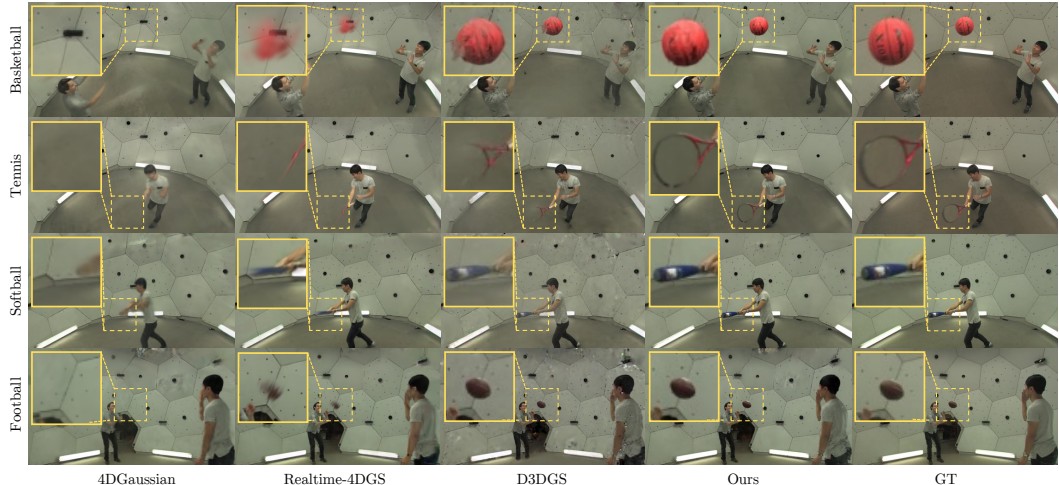

Figure 4: **Visualizations on dynamic sports scenes in the CMU Panoptic Sports dataset.** Compared to prior 4D Gaussian baselines, where fast-moving objects (*e.g.*, ball, bat, and racket) often disappear or become corrupted, our method successfully preserves these objects throughout the sequence, producing more faithful and consistent reconstructions.

A total of 31 cameras were used, and the native resolution of $640 \times 360$ was retained. Evaluation was referenced to four fixed test cameras (IDs of 0, 10, 15, and 30).

**4DGS baselines.** We compare our method with recent 4DGS baselines across various strategies, including (a) explicit parameterizations for 4DGS: Realtime-4DGS (Yang et al., 2024a), 4D-Rotor-Gaussians (Duan et al., 2024)[1], (b) deformable 4DGS: Grid4D (Xu et al., 2024), 4D Gaussian (Wu et al., 2024), MoDec-GS (Kwak et al., 2025), and (c) external supervision: D3DGS (Luiten et al., 2024), TC3DGS (Javed et al., 2024). We note that D3DGS and TC3DGS are designed to handle large inter-frame displacements, such as the Panoptic Sports dataset. All baselines are evaluated following the dynamic scene setups specified in their papers.

## 4.1 EXPERIMENTAL RESULTS ON PANOPTIC SPORTS

In this section, we evaluate our method with various 4DGS baselines on six dynamic sports scenes (Basketball, Boxes, Football, Juggle, Softball, and Tennis) from the CMU Panoptic Sports dataset, which involve rapid human motions and small objects undergoing large inter-frame displacements.

As reported in Table 1, SPIN-4DGS achieves the best PSNR on all six scenes with an average of 30.11 dB, outperforming the strongest explicit baseline Realtime-4DGS (28.38 dB) by +1.73 dB and the external supervision baseline D3DGS (28.70 dB) by +1.41 dB. SSIM also remains consistently high at 0.93, confirming stable reconstruction quality. In particular, our method even surpasses models trained with external supervision such as segmentation maps (*e.g.*, D3DGS, TC3DGS), showing that high fidelity can be achieved without costly priors. Beyond averages, the improvements are most significant on challenging scenes with extreme motion. On Basketball, SPIN-4DGS improves

---

[1]We reimplemented the method using the official PyTorch code released by the authors.

PSNR by over +1.83 dB against the best baseline, D3DGS (Luiten et al., 2024), while on Tennis, where rackets move rapidly and cover large displacements, the margin also surpasses +1.64 dB against the best baseline, Realtime-4DGS (Yang et al., 2024a). Overall, those results highlight the robustness of our approach to challenging fast-motion scenarios where prior methods often fail.

Qualitative comparisons in Figure 4 further illustrate these differences. Deformable baseline (*i.e.*, 4DGaussian, first column) results blur fine-grained structures, explicit Realtime-4DGS (second) results frequently lose moving objects, and even externally supervised (*i.e.*, D3DGS, third) results suffer degradation on small, fast objects like a tennis racket. In contrast, SPIN-4DGS preserves sharp and stable reconstructions across all frames, closely matching the ground truth.

As shown in Figure 4, only the externally supervised baselines (*e.g.*, D3DGS) are able to represent the fast-moving objects (*e.g.*, basketball) in these sports scenes. Although D3DGS achieves a high rendering speed, it requires external supervision during training and incurs substantial storage overhead (1994 MB) to store a large set of Gaussian attributes for the entire sequence. Explicit Realtime-4DGS renders faster than SPIN-4DGS, yet still fails to reconstruct these fast-moving objects and yields lower PSNR. In contrast, SPIN-4DGS attains higher fidelity with a smaller footprint (1261 MB) by storing only Gaussian positions and the network. Such a network-based approach also makes it possible to reconstruct only selected time intervals without storing all Gaussians of the entire sequence. Compared to other network-based deformable baselines, SPIN-4DGS reaches 104 FPS, substantially higher than 4DGaussian (40 FPS) and MoDec-GS (62 FPS), while these methods still struggle to represent fast-moving objects. These observations suggest that SPIN-4DGS offers a practically usable trade-off among fidelity, storage, and speed for fast-motion sports scenes.

## 4.2 ABLATION STUDY AND ANALYSIS

We conduct a series of ablation experiments to demonstrate the proposed method further. First, we analyze the impacts of estimated spatiotemporal position quality and refinement in Table 2 and 3. We also examine a position-reuse setting in Table 4, where positions from existing 4DGS models are provided, to validate the effectiveness of our attributes learning scheme. Finally, we validate the effect of each component of our implicit network in Table 5.

Table 2: **Ablation study on spatiotemporal positions.** We perform ablation studies on (a) varying the early training duration used to estimate spatiotemporal Gaussian positions from the explicit baseline, Realtime-4DGS (Yang et al., 2024a), evaluated without subsequent refinement, and (b) varying the number of frame-wise refinement iterations applied after position estimation.

| (a) Effect of early training duration | | | | | (b) Effect of refinement iterations | | | | |
|---|---|---|---|---|---|---|---|---|---|
| Iteration | PSNR↑ | SSIM↑ | LPIPS↓ | Time↓ | Iteration | PSNR↑ | SSIM↑ | LPIPS↓ | Time↓ |
| 15K | **29.57** | **0.92** | **0.15** | **10m** | 0.5K | 29.86 | 0.92 | 0.14 | 33m |
| 30K | 29.39 | 0.91 | **0.15** | 30m | 1K | 29.89 | 0.92 | 0.14 | 55m |
| | | | | | 2K | 30.05 | 0.92 | 0.14 | 1h 40m |

**Ablation on spatiotemporal positions.** We investigate how the quality of estimated spatiotemporal positions and the amount of refinement affect the final reconstruction. Results are summarized in Table 2, evaluated on the Basketball sequence.

(a) *Effect of early training duration.* Using positions extracted after 15K iterations of Realtime-4DGS (Yang et al., 2024a) already achieves strong performance (29.57 PSNR, 0.92 SSIM) with only 10 minutes of cost. Extending training to 30K iterations not only triples the runtime but also slightly degrades the quality, indicating that long optimization is unnecessary once positions are sufficiently stable.

(b) *Effect of refinement iterations.* We then vary the number of frame-wise refinement steps after position estimation. Increasing refinement from 0.5K to 2K iterations improves PSNR from 29.86 to 30.05, confirming that moderate refinement shows consistent benefits. In practice, we also prune redundant Gaussians during refinement, which reduces background clutter and focuses updates on salient regions, further enhancing efficiency.

**Effects of input position formulation.** We further analyze the effect of spatiotemporal slicing in Table 3 and Figure 5. For a fair comparison, we fix the batch size to 1 and run all experiments on the

football sequence, varying only the position design. We compare two settings: (a) w/o spatiotemporal slicing, where all Gaussians are optimized jointly without slicing, and (b) spatiotemporal slicing (ours), where positions are sliced frame by frame and aligned along the time axis.

Without slicing, as shown in Table 3, Gaussians are optimized jointly in a unified 4D space. In this formulation, a single Gaussian must simultaneously explain multiple timestamps. However, rasterization is performed frame by frame, so optimization that reduces the loss for one frame inevitably makes the Gaussian suboptimal for others. This cross-frame interference weakens the supervision signal, slows convergence, and increases both training time and memory usage. In contrast, spatiotemporal slicing explicitly separates Gaussians by $(x, y, z, t)$ and filters out irrelevant points at each frame. This avoids interference across frames and ensures that optimization is focused on the relevant Gaussians for each time step. As a result, slicing achieves both higher reconstruction fidelity and significantly lower training cost.

Qualitative comparisons in Figure 5 confirm these findings. Under the unified 4D formulation, attributes collapse as Gaussians attempt to describe other timestamps, producing blurred faces and distorted fast-moving objects (*e.g.*, football in the scene). Our sliced formulation decouples Gaussians over time, showing sharper details and temporally consistent reconstructions.

Table 3: **Ablation on spatiotemporal slicing.** We compare a unified 4D formulation (*i.e.*, w/o slicing), where Gaussian positions are optimized jointly across space–time, against our spatiotemporal slicing strategy that assigns positions per frame.

| Spatiotemporal Slicing | PSNR↑ | SSIM↑ | Time↓ | Train Cost↓ |
|:---:|:---:|:---:|:---:|:---:|
| ✗ | 27.48 | 0.89 | 1h 20m | 18GB |
| ✓ | **28.96** | **0.92** | **25m** | **9GB** |

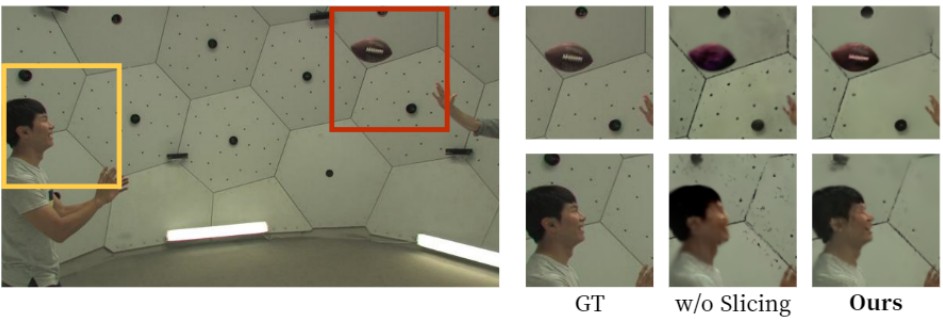

GT     w/o Slicing     **Ours**

Figure 5: **Qualitative comparison of spatiotemporal slicing.** Without slicing, Gaussians simultaneously represent multiple time steps, causing their contributions to overlap and conflict during rasterization. This results in blurred faces and distorted fast-moving objects. Our slicing explicitly separates Gaussians over time, enhancing qualities with temporally consistent reconstructions.

**Impacts of implicit 4DGS scheme.** We further validate the impacts of the proposed implicit 4DGS scheme by reusing positions from strong baselines and retraining only attributes with our framework. In this setting, we extract pre-trained spatiotemporal positions from D3DGS (Luiten et al., 2024) and Realtime-4DGS (Yang et al., 2024a), but discard all their learned attributes. SPIN-4DGS then learns new attributes through its implicit network, while keeping the imported positions learnable to allow for refinement during training. This setup highlights not only the effectiveness of our attribute learning design but also its plug-and-play compatibility with existing 4DGS pipelines, where our scheme consistently enhances reconstruction quality regardless of the underlying position estimator.

As shown in Table 4, SPIN-4DGS consistently improves performance even when initialized with external positions. With D3DGS positions, our method improves PSNR by +2.49 dB and SSIM to 0.95 on the *Softball* scene. Notably, while D3DGS collapses on *Tennis* (28.11/0.91), SPIN-4DGS still achieves 30.51/0.95 with sharp reconstructions. Even with 4DGS positions, our framework still achieves consistent gains overall, demonstrating the effectiveness of the proposed scheme. While SSIM occasionally drops slightly in some cases, we find that this minor loss is easily resolved by our refinement step, showing that performance remains robust and stable.

We find that high-quality positions alone are insufficient for stable reconstructions, as attributes play a critical role in maintaining fidelity under large inter-frame displacements. We also observe that

Table 4: **Compatibility with existing 4DGS baselines.** We reuse pre-trained positions from D3DGS (Luiten et al., 2024) and Realtime-4DGS (Yang et al., 2024a), replacing their attribute optimization with our proposed implicit network training. SPIN-4DGS consistently improves PSNR/SSIM across all scenes, highlighting the compatibility and effectiveness of the proposed implicit 4DGS scheme.

| Method | Basketball | | Boxes | | Football | | Juggle | | Softball | | Tennis | |
|---|---|---|---|---|---|---|---|---|---|---|---|---|
| | PSNR↑ | SSIM↑ | PSNR↑ | SSIM↑ | PSNR↑ | SSIM↑ | PSNR↑ | SSIM↑ | PSNR↑ | SSIM↑ | PSNR↑ | SSIM↑ |
| Realtime-4DGS (Yang et al., 2024a) | 27.89 | **0.92** | 28.17 | **0.93** | 28.35 | **0.93** | 28.68 | **0.93** | 28.67 | **0.93** | 28.50 | **0.93** |
| Realtime-4DGS + Ours | **29.57** | 0.92 | **29.63** | 0.92 | **29.42** | 0.92 | **29.98** | 0.92 | **29.94** | 0.93 | **29.83** | 0.93 |
| D3DGS (Luiten et al., 2024) | 28.22 | 0.91 | 29.46 | 0.91 | 28.49 | 0.91 | 29.48 | 0.92 | 28.43 | 0.91 | 28.11 | 0.91 |
| D3DGS + Ours | **30.50** | **0.94** | **30.26** | **0.94** | **29.21** | **0.94** | **31.04** | **0.95** | **30.92** | **0.95** | **30.51** | **0.95** |

our implicit 4DGS formulation generalizes seamlessly across different baselines, showing strong compatibility and consistent improvements regardless of the source of positions. Taken together, these results highlight SPIN-4DGS as an effective and extensible 4DGS scheme that can enhance a wide range of dynamic scene reconstruction tasks.

Table 5: **Ablation on our implicit network design components.** Starting from the original 4D hash encoder (Chen et al., 2025), we analyze the effects of making positions trainable, applying input position normalization, and changing activation functions. Each modification progressively enhances reconstruction quality. All experiments are performed on the Basketball scene.

| Network Components | | | | Results | | |
|---|---|---|---|---|---|---|
| Trainable position | Normalization | Activation | Hash Map Size | PSNR↑ | SSIM↑ | LPIPS↓ |
| | | ReLU | $2^{21}$ | 18.64 | 0.78 | 0.45 |
| ✓ | | ReLU | $2^{21}$ | 19.17 | 0.78 | 0.46 |
| ✓ | ✓ | ReLU | $2^{21}$ | 29.89 | 0.92 | 0.16 |
| ✓ | ✓ | GELU | $2^{21}$ | 30.05 | 0.92 | 0.14 |
| ✓ | ✓ | GELU | $2^{23}$ | **30.25** | **0.93** | **0.13** |

**Component analysis.** We conduct an ablation study on the design components of our implicit network, summarized in Table 5. Starting from the original 4D hash encoder (Chen et al., 2025) encoder with fixed positions and ReLU activations, reconstruction quality is notably poor (18.64 PSNR, 0.78 SSIM, 0.45 LPIPS). Making positions trainable alone achieves only marginal improvement, indicating that position refinement is insufficient without additional modifications. Applying input position normalization shows a significant gain, boosting PSNR by over +10 dB and reducing LPIPS from 0.45 to 0.16, highlighting its importance for stabilized training. Replacing ReLU with GELU further improves fidelity, enlarging the hash map size further achieves the best overall performances (30.25 PSNR, 0.93 SSIM, 0.13 LPIPS). These results confirm that each component contributes to stability and accuracy, with their combination being crucial for achieving high-quality reconstructions.

## 5 CONCLUSION

In this work, we addressed the challenge of dynamic scene reconstruction under fast motions with large inter-frame displacements. Existing 4DGS methods, including deformable and explicit approaches, often fail to maintain Gaussian attributes in such regimes, resulting in blurred or vanished objects. We introduced SPIN-4DGS, a new framework that learns Gaussian attributes directly from explicit spatiotemporal positions via a feed-forward network, rather than relying on temporal displacement modeling. By decoupling Gaussian position estimation from Gaussian attribute learning, SPIN-4DGS offers a simple yet effective design principle for representing dynamic scenes under fast motions. SPIN-4DGS achieved high-quality reconstructions of fast-moving objects under large inter-frame displacements, as demonstrated through extensive experiments across dynamic sports scenes in the CMU Panoptic Sports dataset. Overall, we believe SPIN-4DGS advances 4D Gaussian Splatting toward practical deployment in challenging real-world scenarios.

ACKNOWLEDGMENT

This work was supported by the National Research Foundation of Korea (NRF) grant funded by the Korea government (MSIT) (RS-2025-24533937) and by the MSIT (Ministry of Science and ICT), Korea, under the Convergence Security Core Talent Training Business Support Program (IITP-2024-RS-2024-00423071), supervised by the IITP (Institute of Information & Communications Technology Planning & Evaluation).

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

# A  MORE ABLATION STUDIES

**Perceptual quality comparison.**    In terms of perceptual quality (LPIPS), SPIN-4DGS also delivers competitive performance across all six scenes on the CMU dataset. As shown in Table 6, our method achieves a mean LPIPS of 0.14, substantially improving over the strongest baseline D3DGS (0.18) while remaining close to the explicit Realtime-4DGS (0.13). These results indicate that SPIN-4DGS effectively suppresses blur and collapse artifacts around fast-moving objects without sacrificing perceptual fidelity, and together with the PSNR/SSIM gains further confirm the robustness of our method under challenging fast-motion scenarios.

Table 6: **Perceptual quality comparison.**  We report LPIPS across six sports scenes on the CMU Panoptic Sports dataset. Lower scores indicate better perceptual quality.

| Model | 4DGS Category | Basketball | Boxes | Football | Juggle | Softball | Tennis | Avg. |
|---|---|---|---|---|---|---|---|---|
| D3DGS (Luiten et al., 2024) | External supervision | 0.18 | 0.17 | 0.19 | 0.15 | 0.19 | 0.17 | 0.18 |
| Realtime-4DGS (Yang et al., 2024a) | Explicit | **0.14** | **0.12** | **0.13** | **0.12** | **0.12** | **0.13** | **0.13** |
| Ours | - | **0.14** | 0.14 | 0.14 | 0.13 | 0.13 | **0.13** | 0.14 |

**Training budget.**    Compared to the strongest baseline D3DGS, SPIN-4DGS offers a better trade-off between reconstruction quality and training time, as summarized in Table 7. With a lightweight 15K-iteration configuration without refinement, SPIN-4DGS already attains 28.82 dB on Basketball, outperforming D3DGS (28.22 dB) while reducing training time from 100 to 45 minutes. Adding a short refinement stage (15K position iterations and 0.5K refinement iterations) further improves PSNR to 29.13 dB in 73 minutes, still faster than D3DGS. With more compute, longer refinement yields additional gains: 15K + 2K reaches 29.51 dB in 142 minutes, and the full 40K + 2K setting achieves 30.05 dB in 195 minutes, *i.e.*, +1.83 dB over D3DGS. Overall, our two-stage design can match or surpass D3DGS with a substantially smaller training budget (even without external supervision), while providing a smooth path to higher fidelity when more budget is available.

Table 7: **Training time breakdown and PSNR on the Basketball scene.**  We compare PSNR, position-estimation time, refinement cost, network training time, and total training time across different SPIN-4DGS configurations and the D3DGS baseline. All time measurements are reported in minutes.

| Method | Setting | PSNR | Position | Refine | Network | Total time |
|---|---|---|---|---|---|---|
| D3DGS | Default | 28.22 | — | — | — | 100 |
| Ours | 15K, No refine | 28.82 | 10 | 0 | 35 | 45 |
| | 15K, Refine (0.5K) | 29.13 | 10 | 33 | 30 | 73 |
| | 15K, Refine (2K) | 29.51 | 10 | 100 | 32 | 142 |
| | 40K, Refine (2K) | 30.05 | 10 | 100 | 85 | 195 |

**Neu3DV benchmark.**    We additionally evaluate SPIN-4DGS on the Neu3DV(Li et al., 2022b) benchmark, which features complex backgrounds and smaller motions. In this experiment, we keep the default SPIN-4DGS training setup without the refinement stage, training only on pre-refinement spatiotemporal positions, with the hash map size hyperparameter set to $2^{23}$ throughout training. As summarized in Table 8, SPIN-4DGS attains the highest average PSNR of 32.19 dB, slightly outperforming the strongest explicit Realtime-4DGS (32.01 dB) by +0.18 dB and clearly improving over deformable methods such as Grid4D (31.49 dB) by +0.70 dB and 4DGaussian (31.01 dB) by +1.18 dB. Moreover, SPIN-4DGS improves over the externally supervised D3DGS baseline (31.04 dB) by 1.15 dB in mean PSNR, and SSIM also remains consistently high at 0.95, matching or surpassing the best competing methods. Qualitative comparisons in Figure 9 show that moving objects (*e.g.*, the tongs and knife) appear blurred or smeared with explicit Realtime-4DGS, whereas SPIN-4DGS reconstructs them with much sharper details, confirming stable reconstruction quality on complex, slower-motion scenes.

**MeetRoom benchmark.**    We further evaluate SPIN-4DGS on the three scenes (Discussion, Trimming, VR Headset) from the MeetRoom benchmark (Li et al., 2022a), which contain cluttered in-

Table 8: **Comparisons on the Neu3DV scenes.** We report PSNR (and SSIM when available) for SPIN-4DGS and baselines across six sequences.

| Method | 4DGS Category | Coffee Martini | | Cook Spinach | | Cut Roasted Beef | | Flame Salmon | | Flame Steak | | Sear Steak | | Avg. | |
|---|---|---|---|---|---|---|---|---|---|---|---|---|---|---|---|
| | | PSNR↑ | SSIM↑ | PSNR↑ | SSIM↑ | PSNR↑ | SSIM↑ | PSNR↑ | SSIM↑ | PSNR↑ | SSIM↑ | PSNR↑ | SSIM↑ | PSNR↑ | SSIM↑ |
| Grid4D (Xu et al., 2024) | Deformable | 28.30 | 0.90 | 32.58 | 0.95 | 33.22 | 0.95 | 29.12 | 0.91 | 32.56 | **0.96** | 33.16 | **0.96** | 31.49 | 0.94 |
| 4DGaussian (Wu et al., 2024) | Deformable | 27.34 | 0.90 | 32.50 | 0.94 | 32.26 | 0.94 | 27.99 | 0.90 | 32.54 | 0.95 | 33.44 | 0.95 | 31.01 | 0.93 |
| D3DGS (Luiten et al., 2024) | External supervision | 27.32 | - | 32.97 | - | 31.75 | - | 27.26 | - | 33.24 | - | 33.68 | - | 31.04 | - |
| Realtime-4DGS (Yang et al., 2024a) | Explicit | 28.33 | - | 32.93 | - | 33.85 | - | **29.38** | - | **34.03** | - | 33.51 | - | 32.01 | - |
| Realtime-4DGS(re-impl.) (Lee et al., 2025) | Explicit | 27.92 | **0.92** | 33.58 | **0.96** | 33.96 | **0.96** | 28.72 | **0.92** | 33.96 | **0.96** | 33.61 | **0.96** | 31.96 | **0.95** |
| Ours | - | **28.42** | **0.92** | **33.61** | **0.96** | **34.05** | **0.96** | 28.97 | **0.92** | 33.96 | **0.96** | **34.14** | **0.96** | **32.19** | **0.95** |

door backgrounds and relatively small motions. Using the same configuration as in our main experiments, we train SPIN-4DGS only on the pre-refinement spatiotemporal positions, with the hashmap size fixed to $2^{23}$ throughout training. Table 9 shows that SPIN-4DGS consistently achieves higher PSNR than all baselines, including StreamRF (26.72 dB), 3DGStream (30.79 dB), and explicit Realtime-4DGS (30.47 dB). Overall, it provides an average improvement of +1.6 dB over the explicit Realtime-4DGS. In particular, on the Trimming scene, SPIN-4DGS attains 32.41 dB, improving over the explicit Realtime-4DGS (30.16 dB) by more than 2 dB, while also maintaining a high SSIM of 0.96. Qualitative comparisons in Figure 10 illustrate these effects: while Realtime-4DGS often fails to faithfully reconstruct objects due to noise and artifacts in both the foreground and background, SPIN-4DGS yields sharper object reconstructions and substantially cleaner backgrounds. These results suggest that our model remains robust even in indoor scenes with cluttered backgrounds and relatively small motions.

Table 9: **Comparisons on the MeetRoom scenes.** We report PSNR and SSIM for SPIN-4DGS and baselines across three sequences and their average. Note that only average PSNR scores are available for StreamRF and 3DGStream.

| Method | Category | Discussion | | Trimming | | VR Headset | | AVG | |
|---|---|---|---|---|---|---|---|---|---|
| | | PSNR↑ | SSIM↑ | PSNR↑ | SSIM↑ | PSNR↑ | SSIM↑ | PSNR↑ | SSIM↑ |
| StreamRF (Li et al., 2022a) | Implicit | – | – | – | – | – | – | 26.72 | – |
| 3DGStream (Sun et al., 2024) | Explicit | – | – | – | – | – | – | 30.79 | – |
| Realtime-4DGS (Yang et al., 2024a) | Explicit | 31.51 | **0.96** | 30.16 | 0.94 | 29.74 | **0.95** | 30.47 | 0.95 |
| Ours | - | **32.30** | **0.96** | **32.41** | **0.96** | **31.42** | **0.95** | **32.04** | **0.96** |

**Comparison to frame-wise 3DGS.** To verify the benefits of our temporally shared implicit representation over frame-wise 3DGS (*i.e.*, a collection of 3DGS on each frame), we directly compared SPIN-4DGS with frame-wise 3DGS (Kerbl et al., 2023) and D3DGS (Luiten et al., 2024) on the CMU Panoptic Sports benchmark, where we referred to the results reported in the D3DGS. Table 10 shows that SPIN-4DGS achieves the best performance on all six CMU Panoptic Sports scenes, surpassing both frame-wise 3DGS and D3DGS. Quantitatively, SPIN-4DGS improves PSNR by 1.9dB on average frame-wise 3DGS and with gains of up to +3.03dB on *Basketball*. These substantial gains suggest that our temporally shared implicit field can better exploit temporal regularities and cross-frame correlations, yielding a more coherent 4D representation than a set of independently optimized frame-wise 3DGS models.

# B    MORE QUALITATIVE RESULTS

**Qualitative results on spatiotemporal position refinement.** Figure 6 visualizes the ablation study on the position refinement in Table 2. With only 0.5K refinement iterations, SPIN-4DGS already captures stable object structures without collapse, while increasing iterations to 1K and 2K further improves fine details such as ball edges and racket nets. Specifically, our refinement stage employs Gaussian densification, which both allocates new Gaussians in regions that lack detail and prunes redundant ones, incrementally filling in missing structures while reducing the number of unnecessary Gaussians. This process can alleviate the dependence on initially collected positions, allowing SPIN-4DGS to train on a reliable input point set. These results indicate that SPIN-4DGS is effective even with minimal refinement budgets and that additional iterations mainly serve to sharpen fine details rather than prevent collapse.

Table 10: **Comparison with frame-wise 3DGS on the CMU Panoptic Sports benchmark.** We report PSNR across six sports, showing that SPIN-4DGS consistently outperforms both 3DGS and D3DGS across all sequences.

| Model | Basketball | Boxes | Football | Juggle | Softball | Tennis | Avg. |
|---|---|---|---|---|---|---|---|
| 3DGS (Kerbl et al., 2023) | 27.02 | 28.74 | 28.49 | 28.19 | 28.77 | 28.03 | 28.21 |
| D3DGS (Luiten et al., 2024) | 28.22 | 29.46 | 28.49 | 29.48 | 28.43 | 28.11 | 28.70 |
| Ours | **30.05** | **29.91** | **29.99** | **30.31** | **30.24** | **30.14** | **30.11** |

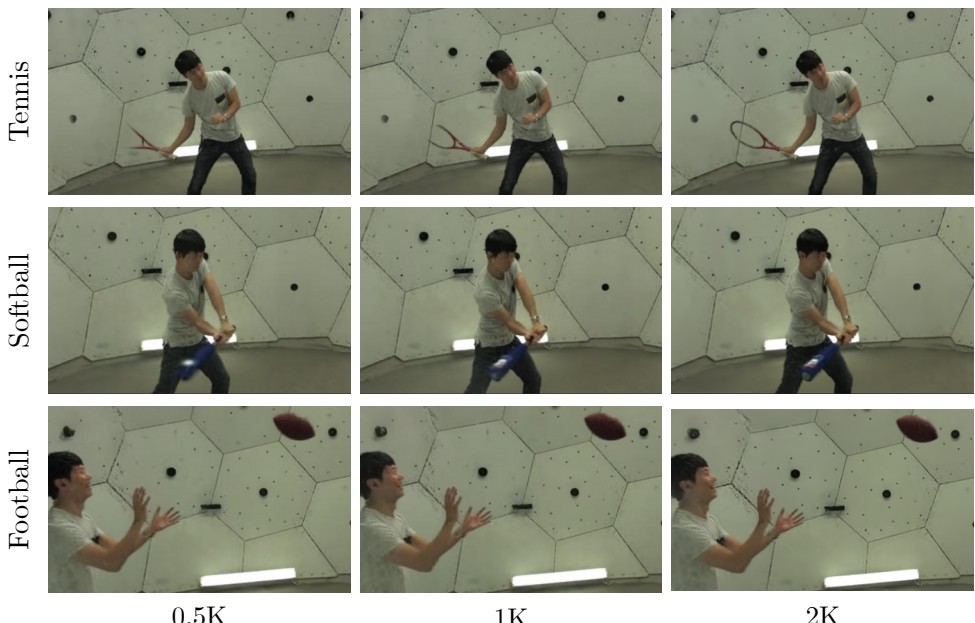

Figure 6: **Visualizations on spatiotemporal position refinement.** We visualize the effect of varying refinement iterations (0.5K, 1K, 2K) corresponding to Table 2b. Even with only 0.5K iterations, SPIN-4DGS retains consistent object structures without collapse, while progressively increasing the number of iterations recovers fine details such as ball edges and racket nets.

**Qualitative results with various baselines.** Figure 7 shows qualitative comparisons on the CMU Panoptic Sports dataset for two baseline methods in Table 1: MoDec-GS and 4D-Rotor-Gaussians. For MoDec-GS, while the background remains relatively stable, the player becomes heavily blurred and fast-moving objects (*e.g.*, basketball, tennis racket) are often not reconstructed correctly, similar to prior deformable 4DGS approaches. 4D-Rotor-Gaussians retains these objects more reliably, but still exhibits noticeable blur and geometry instability of the player under fast motion. In contrast, SPIN-4DGS reconstructs both the moving objects and the player with sharp and temporally stable geometry, yielding visually cleaner and more consistent results than both baselines.

**Qualitative results on compatibility with existing 4DGS baselines.** Figure 8 presents qualitative results for the compatibility study in Table 4. Across scenes, our method reconstructs stable backgrounds and more temporally consistent details, while the original baselines often blur or lose fast-moving objects. These results confirm that the proposed scheme integrates effectively with existing 4DGS approaches and potentially improves their reconstruction fidelity under large inter-frame displacements.

## C  DISCUSSIONS

**Recent trends in 4DGS.** Recent concurrent 4DGS methods (Dai et al., 2025; Oh et al., 2025; Lyu et al., 2025) have been introduced, often sharing similar architectural components but pursuing different goals and design constraints than ours. 4DGV (Dai et al., 2025) focuses on reducing model

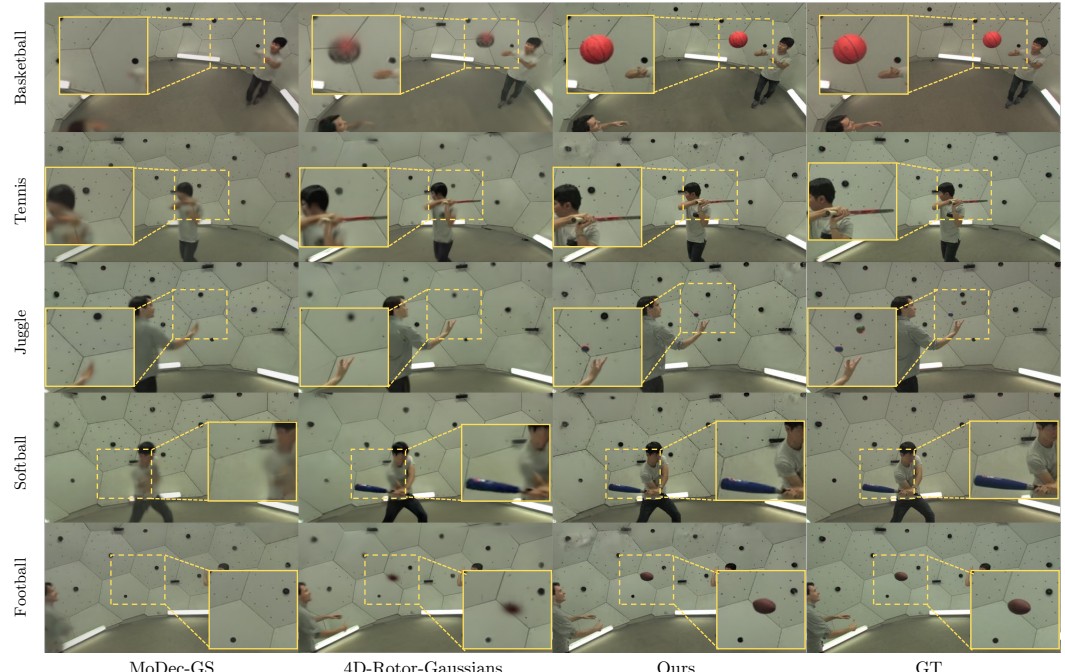

Figure 7: **Visualization on dynamic sports scenes in the CMU Panoptic Sports dataset.** We visualize the methods evaluated in Table 1, including MoDec-GS and 4D-Rotor-Gaussians. These 4DGS baselines often exhibit geometry instability and fast-motion artifacts, whereas SPIN-4DGS preserves the object and player geometry sharply throughout the sequence.

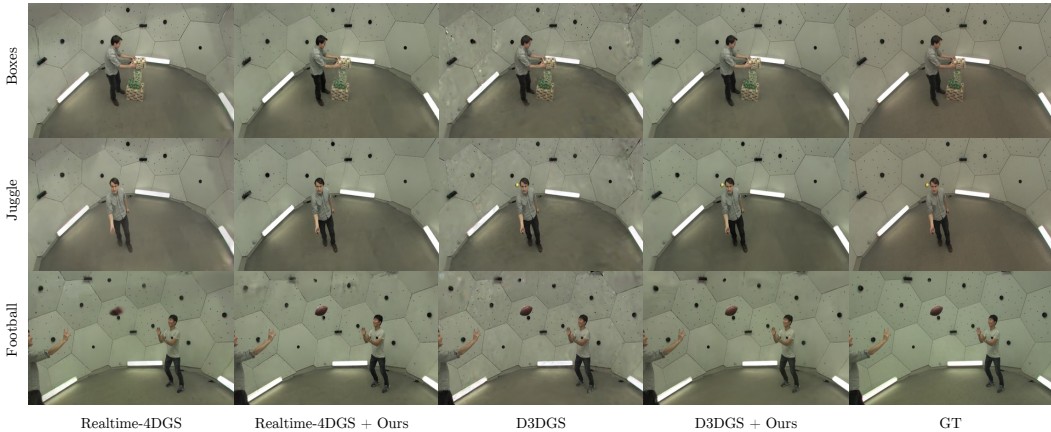

Figure 8: **Visualizations on compatibility with existing 4DGS baselines.** We visualize the results of compatibility ablation study in Table 4, where pre-trained positions from 4DGS and D3DGS are reused while attributes are retrained with SPIN-4DGS. Our method consistently produces clear and more stable reconstructions than the original baselines.

size for complex dynamic scenes by leveraging external 2D segmentation, thereby relying on external supervision. Hybrid 3D–4DGS (Oh et al., 2025) builds upon 4DGS and mitigates redundancy in static regions by converting Gaussians that barely change over time into 3D ones; however, dynamic Gaussians are still shared across multiple frames, leaving cross-frame interference unresolved for fast motions. SCas4D (Lyu et al., 2025), built on Dynamic3DGS, accelerates online 4D reconstruction via cluster-wise shared transformations, but this design couples all Gaussians within a cluster to a single motion model and makes the method sensitive to the initial K-means clustering. By contrast, SPIN-4DGS directly predicts frame-wise attributes from explicit spatiotemporal positions

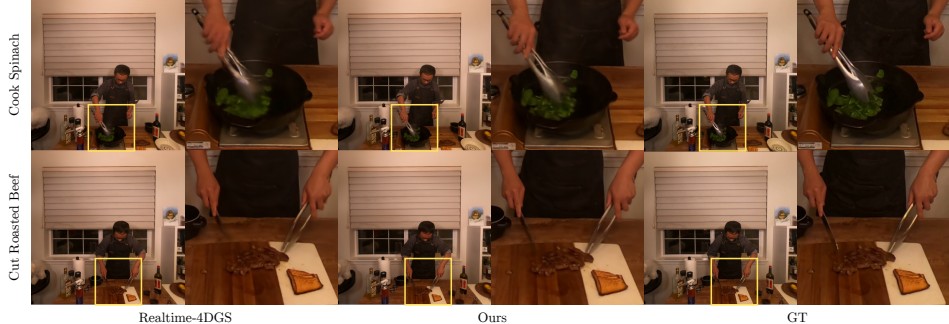

Figure 9: **Qualitative comparison on the Neu3DV benchmark.** We visualize representative results from two Neu3DV scenes (*e.g.*, Cook Spinach and Cut Roasted Beef) reported in Table 8. 4DGS blurs the boundaries of moving objects (*e.g.*, tongs and knife), whereas SPIN-4DGS achieves significantly cleaner and higher-fidelity reconstructions.

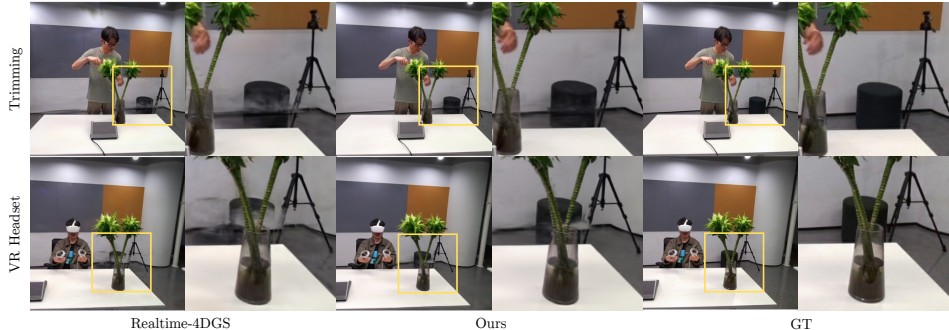

Figure 10: **Qualitative comparison on the MeetRoom benchmark.** We visualize representative results from two MeetRoom scenes (*e.g.*, Trimming and VR Headset) reported in Table 9. The 4DGS exhibit noticeable background noise and often fail to reconstruct the foreground objects faithfully, whereas SPIN-4DGS achieves sharper object reconstruction with substantially cleaner backgrounds.

with an implicit decoder, avoiding cross-frame interference and enabling stable reconstruction of fast motions without requiring external supervision or cluster-based initialization.

**Key differences from prior 4DGS.** The key conceptual difference between SPIN-4DGS and prior 4DGS methods is how temporal appearance is parameterized. SPIN-4DGS rasterizes each frame from explicit spatiotemporal positions and predicts Gaussian attributes as a function of the spatiotemporal coordinates via $f_\theta(\mathbf{u}, t)$. This design keeps positions explicit, allows only light per-sample adjustments during training, and decouples attribute optimization across timesteps, so different samples can receive different attributes without directly sharing a single attribute vector over time.

In contrast, standard explicit 4DGS (*e.g.*, Realtime-4DGS (Yang et al., 2024a), 4D-Rotor-Gaussians (Duan et al., 2024)) generate each frame by time-slicing a 4D primitive, and deformable 4DGS (*e.g.*, 4DGaussian (Wu et al., 2024), Grid4D (Xu et al., 2024)) transform canonical Gaussians over time; in both cases, multiple timesteps share the same underlying Gaussian attributes. As a result, a single set of Gaussian attributes (or transforms) simultaneously influences multiple frames, and updates driven by one frame can easily degrade others under large inter-frame displacements, leading to temporal collapse of fast-moving objects. Our temporal parameterization breaks this coupling at the attribute level and instead places the temporal representation in $f_\theta$ rather than in a pointwise deformation field. As shown in Table 1, this leads to more stable training under large inter-frame displacements and yields higher reconstruction quality (PSNR/SSIM) on fast-motion scenes.

Moreover, several deformable 4DGS methods deliberately avoid time-varying color and opacity, as allowing these attributes to change over time often produces unstable or implausible surfaces in novel views and makes tracking difficult. By predicting color and opacity from spatiotemporal coordinates through $f_\theta$ while keeping the underlying Gaussian support stable, SPIN-4DGS can model genuinely time-varying appearance without sacrificing temporal stability, which is crucial for fast-moving objects with large inter-frame displacements.

**Limitations.** We empirically observe that our method is most effective when a sufficiently dense set of spatiotemporal positions is available. In large-scale outdoor or highly cluttered scenes with sparse initial positions, achieving high reconstruction quality may require longer refinement (densify/prune) schedules to allocate more Gaussians, thereby increasing the overall training cost. In addition, for small objects or regions with very few initial points, we empirically observe that even multiple refinement iterations struggle to generate or recover sufficient missing points.

**LLM policy.** LLMs were used only for editing (grammar and typographical errors) and did not contribute to idea generation, analysis, experimental design, or interpretation.

