# OpenReview forum: "Implicit 4D Gaussian Splatting for Fast Motion with Large Inter-Frame Displacements"
_ICLR.cc/2026/Conference — ICLR 2026 Poster_

### Official Review · Reviewer_H2vU · 2025-10-20

**Soundness:** 3
**Presentation:** 3
**Contribution:** 2
**Rating:** 4
**Confidence:** 3

**Summary:**

The paper presents SPIN-4DGS, a new 4D Gaussian Splatting (4DGS) method designed to handle scenes with fast-moving objects, which typically degrade reconstruction quality and cause motion blur or object disappearance. Instead of employing a time-varying, deformable formulation, SPIN-4DGS follows an explicit strategy and first constructs Gaussian sets independently per frame, then employs a lightweight neural network to refine latent features in a frame-wise manner, which are then fed into implicit decoder to predict Gaussian attributes. Experiments on the CMU Panoptic dataset, which includes sports sequences with fast motion (e.g., moving balls and players), demonstrate improved reconstruction quality over tested methods in terms of PSNR and SSIM.

**Strengths:**

- The quantitative and qualitative results show clear improvements over existing dynamic Gaussian splatting methods, including 4DGS and others that rely on depth supervision or deformable modeling.
- The paper offers thorough ablation studies, systematically analyzing the contributions of different components such as training iterations, refinement steps, and network design choices.
- The proposed refinement module is modular and lightweight, making it a potential plug-and-play replacement for attribute optimization stages in existing 4DGS pipelines to enhance reconstruction performance in both PSNR and SSIM metrics.
- The approach is conceptually simple yet effective, avoiding complex temporal modeling while still capturing inter-frame consistency through learned refinements.

**Weaknesses:**

- The paper briefly mentions other explicit 4DGS approaches, such as 4DGS (Yang et al., 2023) and 4D-Rotor-Gaussians (Duan et al., 2024), but lacks an in-depth comparison or discussion of their relative advantages and limitations. In particular, results for 4D-Rotor-Gaussians are missing from the comparison table, which limits the reader’s ability to judge empirical progress.
- The method is evaluated solely on the CMU Panoptic dataset, which features controlled indoor scenes with limited background complexity. Evaluation on more diverse benchmarks (e.g., N3DV (Li et al., CVPR 2022) and D-NeRF (Pumarola et al., 2021)), commonly used in prior dynamic scene reconstruction works, would better establish the generalizability of the proposed approach.
- The novelty claim could be articulated more clearly. The proposed frame-wise construction and refinement pipeline resembles modular post-processing strategies from prior 4DGS extensions. A more explicit positioning of the technical contribution relative to these prior methods would help.

Minor Issues
- Confirm whether cited preprints were published at major conferences (e.g., Yang et al., 2023 appeared at ICLR 2024).
- Line #177: should read f<sub>θ</sub> instead of F<sub>θ</sub>.

**References**
- Li et al. Neural 3D Video Synthesis from Multi-View Video. CVPR 2022.
- Pumarola et al. D-NeRF: Neural Radiance Fields for Dynamic Scenes. CVPR 2021.

**Questions:**

1. Could you elaborate on the key conceptual differences between SPIN-4DGS and other explicit 4DGS-based methods (e.g., 4DGS, 4D-Rotor-Gaussians)? Specifically, how does your per-frame refinement compare to time-slicing or deformable formulations in terms of accuracy and efficiency?
2. Please include quantitative comparisons against 4D-Rotor-Gaussians (Duan et al., 2024) to provide a more complete empirical evaluation.
3. To assess the generalizability of the proposed method, it would be valuable to add experiments on N3DV and D-NeRF datasets or discuss potential limitations when applied to more complex outdoor or cluttered environments.

---

> ### Author Response · Authors · 2025-11-20
>
> Dear reviewer H2vU,
>
> Thank you for noting our empirical gains, thorough ablations, and the practicality of the refinement. We will address your concerns and questions in the response below.
>
> ---
>
> ### [W1 & Q1] Conceptual differences from explicit 4DGS and deformable formulations
>
> ---
>
> The key conceptual differences between SPIN-4DGS and prior works is that SPIN‑4DGS rasterizes each frame from explicit spatiotemporal positions (allowing small gradient update during training) while predicting attributes as a function of $(x,y,z,t)$.  This decouples attribute optimization across timesteps and reduces cross‑frame interference that arises when a single set of Gaussian attributes influences multiple frames. In contrast, explicit 4DGS methods (e.g., 4DGS, 4D-Rotor-Gaussians) time‑slices a 4D primitive and deformable methods transform canonical Gaussians over time, both of which couple frames through shared attributes or transforms, making them fragile under large inter‑frame displacements.  Consequently, we found that SPIN‑4DGS achieves higher reconstruction quality, as reflected by improved PSNR/SSIM on fast-motion scenes. We will include the discussions in the revised manuscript.
>
> ----
>
>
> ### [W1 & Q2] Quantitative comparison with 4D-Rotor-Gaussians
>
> ---
>
> We provide 4D-Rotor-Gaussians results with the existing comparison on the CMU dataset in the table below. Compared to 4D-Rotor-Gaussians, SPIN-4DGS achieves consistently higher reconstruction quality on the CMU Panoptic Sports dataset, with an average PSNR gain of about 3.0 dB, corresponding to roughly an 11% relative improvement. Furthermore, we observe that 4D-Rotor-Gaussians lag behind the 4DGS baselines by about 1.2 dB on average (27.16 dB vs. 28.38 dB). Qualitatively, while 4D-Rotor-Gaussians can reasonably reconstruct simple objects such as the ball, we consistently observe broken or unstable geometry for more subtly moving body parts, leading to visibly degraded frames. We will add this in the revised manuscript.
>
> | Model           | Basketball | Boxes | Football | Juggle | Softball | Tennis | Mean  |
> |-----------------|------------|-------|----------|--------|----------|--------|-------|
> | D3DGS           | 28.22      | 29.46 | 28.49    | 29.48  | 28.43    | 28.11  | 28.70 |
> | 4DGS            | 27.89      | 28.17 | 28.35    | 28.68  | 28.67    | 28.50  | 28.38 |
> | 4D-Rotor-Gaussians  | 27.76      | 26.94 | 26.54    | 27.62  | 27.03    | 27.09  | 27.16 |
> | SPIN-4DGS (Ours) | 30.05      | 29.91 | 29.99    | 30.31  | 30.24    | 30.14  | 30.11 |
>
> ----
>
>
> ### [W2 & Q3] Quantitative evaluation of generalizability on Neu3DV and discussion of potential limitations
>
> ---
>
> #### Generalizability on Neu3DV
>
> We additionally evaluate on the Neu3DV benchmark, which features complex backgrounds and slow motions. SPIN‑4DGS attains a mean PSNR of 32.19 dB, improving over 4DGS (32.00 dB) by +0.19 dB and over D3DGS (31.04 dB) by +1.15 dB. This indicates that our approach remains competitive even when inter-frame displacements are small and backgrounds are cluttered, while D3DGS (the strongest baseline on CMU dataset) even performs worse than the explicit 4DGS baseline. We will add Neu3DV results in the revised manuscript.
>
> | Model | 4DGS Category        | Mean PSNR |
> |-------|----------------------|-----------|
> | D3DGS | External supervision | 31.04     |
> | 4DGS  | Explicit             | 32.00     |
> | SPIN-4DGS (Ours)  | -                    | 32.19     |
>
> ---
> #### Discussion of potential limitations
>
> We empirically found that our method is most effective when a sufficiently dense set of spatiotemporal positions is available. In large‑scale outdoor or highly cluttered scenes with sparse initial positions, achieving high quality may require longer refinement (densify/prune) to allocate more Gaussians, which increases training cost. We will clarify these results and limitations in the revised manuscript.

---

> ### Author Response · Authors · 2025-11-20
>
> ### [W3] Positioning the technical contribution of SPIN-4DGS
>
> ---
>
> Thank you for your valuable suggestion. We would like to emphasize that SPIN‑4DGS is not just an extension of 4DGS in the sense of adding a simple post-processing module, nor does it reuse Gaussian parameters from 4DGS. Instead, 4DGS (or other deformable methods) is used purely as a position estimator to collect spatiotemporal point locations. Conceptually, our pipeline takes these positions as input and reconstructs all remaining Gaussian attributes. In this sense, the main technical contribution of SPIN‑4DGS is to formulate an explicit 4D Gaussian parameter estimator conditioned on trainable spatio‑temporal positions, rather than a modular post-processing step operating on already optimized Gaussians. In contrast to such post-processing-style strategies, SPIN‑4DGS does not operate on fixed Gaussian attributes from a pre-trained 4DGS model; we only use 4DGS-like methods to estimate initial spatiotemporal positions and then learn all Gaussian attributes from scratch on learnable (not frozen) spatiotemporal positions.
>
> To implement this estimator in practice, we design an implicit 4DGS decoder whose goal is to reliably generate all Gaussian parameters from explicit positions. Because each attribute (scale, rotation, opacity, color) lies in a different numerical regime and optimization landscape (e.g., rotation on a manifold, scale/opacity on positive reals, and color parameterized with spherical harmonics), naively training a single decoder architecture with a shared optimization configuration across all outputs leads to an ill-conditioned and unstable training problem.  To address this, we design attribute-aware decoder components and optimization schemes, where carefully chosen activation functions, learning rates, weight initialization, and bias initialization are crucial for robustly generating high-quality Gaussian parameters. We also note that prior deformable 4DGS methods often keep Gaussian color and opacity time-invariant for stability, as allowing these attributes to vary over time tends to introduce inconsistent surfaces in novel views. In contrast, SPIN‑4DGS explicitly predicts time-varying color and opacity from $(x,y,z,t)$, and our attribute-aware decoder design is key to maintaining stability under large inter-frame displacements, providing a solid foundation for more advanced architectures and training strategies.
>
> ---
>
> ### Minor issues
>
> ---
>
> Thanks for your editorial comments. We will revise them in the final draft.
>
> ---

---

> > ### Comment · Reviewer_H2vU · 2025-11-26
> >
> > Thank you for the detailed and constructive responses. The additional quantitative comparisons against 4D-Rotor-Gaussians, the Neu3DV results, and the extended discussion on conceptual differences and limitations are all helpful and address several of my earlier concerns. Your clarifications regarding the technical contribution and the role of the attribute-aware decoder are also appreciated.
> >
> > However, I would like to note that the current rebuttal does not appear to be accompanied by a revised manuscript. Since ICLR allows updated versions during the author response period, it is important for the final assessment that the newly added results (particularly the Neu3DV evaluation and the qualitative comparisons with 4D-Rotor-Gaussians) and expanded discussions are actually reflected in the paper itself.
> >
> > Before I can fully judge the completeness and clarity of the submission, I would need to see:
> > 1. **The new quantitative tables** (including 4D-Rotor-Gaussians and Neu3DV) integrated into the main text or appendix.
> > 2. **Qualitative visualizations** demonstrating the claimed improvements, especially on the failure cases of baselines (e.g., geometry instability, fast-motion artifacts).
> > 3. **The expanded discussion** on conceptual differences, generalizability, and limitations incorporated into the revised sections of the paper.
> >
> > These additions are important to evaluate the overall contribution and to ensure that the improvements outlined in the rebuttal are transparently documented.
> >
> > I look forward to reviewing the updated manuscript with these elements included.

---

> ### Author Response · Authors · 2025-11-28
>
> Dear reviewer H2vU,
>
>
> We appreciate your efforts in reviewing our paper and actively participating in the discussion.
>
> ---
> We would like to clarify that most issues you raised were already addressed in the revised manuscript and included in the PDF that we uploaded a week ago (on 22 Nov). It seems that some of the changes may not have been easily visible, possibly because the color highlighting was not applied in that version. To avoid any confusion for you and for other reviewers/AC, we have now re-uploaded the revised PDF with highlighted in blue.
>
> This highlighted version specifically includes the quantitative results for 4D-Rotor-Gaussians in Table 1, generalizability results on the Neu3DV benchmark in Table 8 (Appendix), new qualitative visualizations in Figure 7 (Appendix), and the extended discussion on conceptual differences and limitations in the Appendix.
>
> We apologize for any inconvenience the previous version may have caused and appreciate your careful attention to our submission.
> Please let us know if any further clarification or additional revisions are needed. We will be glad to address them promptly.
>
> Thank you once again for your thoughtful and constructive review.

---

### Official Review · Reviewer_B2ix · 2025-10-23

**Soundness:** 2
**Presentation:** 3
**Contribution:** 3
**Rating:** 6
**Confidence:** 3

**Summary:**

The paper proposes a novel approach for dynamic scene reconstruction, with a particular focus on handling fast object motions. The proposed model, SPIN-4DGS, predicts Gaussian attributes directly from spatiotemporal positions using a lightweight network that learns shared representations across Gaussians. Unlike conventional methods that store attributes explicitly for each Gaussian, SPIN-4DGS encodes them implicitly within the network parameters, thereby significantly improving memory efficiency. Experimental results show that the proposed approach achieves substantially higher accuracy than existing methods, especially in scenes involving rapid motion.

**Strengths:**

- The paper is well written and easy to follow, clearly articulating its motivation and main contributions.
- It introduces a lightweight yet effective framework for dynamic scene reconstruction that performs robustly in scenarios involving fast object motions.
- The idea of predicting Gaussian attributes directly from spatiotemporal positions is novel and well-motivated, leading to more reliable attribute learning for fast-moving objects with large displacements.
- The paper provides comprehensive ablation studies and analyses that effectively demonstrate the impact of the estimated spatiotemporal positions and the proposed attribute learning mechanism on overall performance.

**Weaknesses:**

- Experiments are conducted only on sequences from the Panoptic dataset, which limits the generalizability of the results. Additional evaluations on some other challenging benchmarks with complex motions such as D-NeRF, Neu3DV, MeetRoom, and DeskGames would strengthen the empirical validation.
- The paper does not include comparisons with the recently proposed MoDec-GS model, which is a closely related approach and an important baseline for dynamic scene reconstruction.
- The evaluation in Table 1 relies solely on distortion-based metrics (PSNR and SSIM) and does not report perceptual quality measures such as LPIPS, which are essential for a more comprehensive assessment.
- Several recent related works including 4DGV [1], 3D-4DGS [2], and SCas4D [3] are missing from the related work discussion and could provide valuable context and comparison for positioning the proposed method.

     [1] Dai et al., 4D Gaussian Videos with Motion Layering, ACM TOG 2025

     [2] Oh et al., Hybrid 3D-4D Gaussian Splatting for Fast Dynamic Scene Representation, arXiv 2025

     [3] Lyu et al., SCas4D: Structural Cascaded Optimization for Boosting Persistent 4D Novel View Synthesis, TMLR 2025

- The authors did not discuss any limitation of the suggested approach? For instance, could the authors provide results or discussion on the model’s performance in static or near-static scenes, where inter-frame displacements are minimal?

**Questions:**

See weaknesses.

---

> ### Author Response · Authors · 2025-11-20
>
> Dear reviewer B2ix,
>
> We appreciate your positive assessment of the paper’s clarity, lightweight design, and the idea of predicting Gaussian attributes directly from spatiotemporal positions. We will address your concerns and questions in the response below.
>
> -----
>
> ### [W1] Additional benchmarking experiments
>
> ---
>
> We would like to emphasize that our primary application is fast-moving objects with large displacements (fast sports motion), such as the CMU dataset, which directly exposes the failure mode we analyze. To broaden coverage, we additionally evaluate on Neu3DV, which features complex backgrounds with smaller inter‑frame displacements. SPIN‑4DGS achieves a mean PSNR of 32.19 dB, improving over explicit 4DGS (32.00 dB) by 0.19 dB and over D3DGS (31.04 dB) by 1.15 dB. This suggests that our approach remains competitive even beyond the large‑motion regime, while D3DGS (the strongest baseline on CMU) performs worse than the explicit 4DGS baseline. We will include the Neu3DV results in the revised manuscript.
>
> | Model | 4DGS Category        | Mean PSNR |
> |-------|----------------------|-----------|
> | D3DGS | External supervision | 31.04     |
> | 4DGS  | Explicit             | 32.00     |
> | SPIN-4DGS (Ours)  | -                    | 32.19     |
>
> ---
>
> ### [W2] Comparison to the MoDec-GS
>
> ---
>
> We thank the reviewer for sharing the recent baseline. We report an additional comparison on the Basketball scene in the Panoptic dataset in the table below. Specifically, SPIN-4DGS achieves 30.05 dB PSNR, outperforming MoDec-GS (27.42 dB) by 2.63 dB.  Although MoDec-GS is designed by handling complex real-world motions with a deformable formulation, we observed that it still fails to robustly reconstruct fast-moving objects in this scenario and exhibits notable artifact collapse. In addition, we note that MoDec-GS runs at 68 FPS on this scene, whereas SPIN-4DGS reaches 114 FPS (roughly 1.7× faster). These results suggest that the existing deformable representations alone are insufficient for such challenging sports-motion scenarios, whereas our design provides a practically effective alternative. We will add this in the revised manuscript.
>
> | Method                   | 4DGS Category | PSNR  | SSIM | LPIPS |
> |--------------------------|---------------|-------|------|-------|
> | 4DGaussians  | Deformable    | 27.28 | 0.90 | 0.16  |
> | MoDec-GS                 | Deformable    | 27.42 | 0.90 | 0.15  |
> | 4DGS                     | Explicit      | 27.89 | 0.92 | 0.14  |
> | SPIN-4DGS (Ours)                | -             | 30.05 | 0.92 | 0.14  |
>
>
> ----
>
> ### [W3] Quality metrics LPIPS
>
> ---
>
>  We thank the reviewer for this helpful suggestion and have added LPIPS as a perceptual metric in the table below. On the Panoptic dataset, our method reduces the mean LPIPS from 0.18 (D3DGS) to 0.14, while remaining comparable to 4DGS (0.13). These results confirm that our approach is also competitive in terms of perceptual quality, and we will include the LPIPS scores and corresponding discussion in the revised manuscript.
>
>
> | Model | 4DGS Category        | Basketball | Boxes | Football | Juggle | Softball | Tennis | Mean |
> |-------|----------------------|-----------:|------:|---------:|-------:|---------:|-------:|-----:|
> | D3DGS | External supervision | 0.18       | 0.17  | 0.19     | 0.15   | 0.19     | 0.17   | 0.18 |
> | 4DGS  | Explicit             | 0.14       | 0.12  | 0.13     | 0.12   | 0.12     | 0.13   | 0.13 |
> | SPIN-4DGS (Ours)  | -                    | 0.14       | 0.14  | 0.14     | 0.13   | 0.13     | 0.13   | 0.14 |
>
> ---

---

> ### Author Response · Authors · 2025-11-20
>
> ### [W4] Comparison to concurrent 4DGS baseline
>
> ---
> We will add a related‑work paragraph covering these concurrent efforts and clarify that their goals and design challenges are quite different from ours. First, 4DGV[1] focuses on reducing the model size while handling complex dynamic scenes by leveraging external 2D segmentation. In contrast, SPIN-4DGS does not rely on such external supervision. Hybrid 3D–4DGS[2] builds upon 4DGS and focuses on reducing redundancy in static regions by converting Gaussians that barely change over time into 3D ones. However, dynamic Gaussians are still shared across multiple frames, which leaves cross-frame interference for fast motions. SCas4D [3], built on Dynamic3DGS, accelerates online 4D reconstruction via cluster-wise shared transformations, but this design couples all Gaussians in a cluster to a single motion model and makes the method sensitive to the initial K-means clustering, whereas SPIN-4DGS directly predicts per-Gaussian, per-frame attributes with an implicit decoder without any cluster-based initialization. Nevertheless, we will include the discussion in the final draft.
>
> [1] Dai et al., 4D Gaussian Videos with Motion Layering, ACM TOG 2025
>
> [2] Oh et al., Hybrid 3D-4D Gaussian Splatting for Fast Dynamic Scene Representation, arXiv 2025
>
> [3] Lyu et al., SCas4D: Structural Cascaded Optimization for Boosting Persistent 4D Novel View Synthesis, TMLR 2025
>
> ----
>
> ### [W5-1] The authors did not discuss any limitation of the suggested approach?
>
> ----
>
> One potential limitation is that it depends on the initial point estimation, so objects or regions not captured at this stage cannot be fully recovered in later steps. We note that it is a common issue shared by prior methods, including deformable and explicit 4DGS baselines, all of which depend on the quality of the initial SfM point estimation. To alleviate this reliance on the initial points, we proposed a refinement stage that post-processes regions with inaccurate or insufficiently represented initial positions. As shown in the tennis example in Fig. 6, the racket is partially missing with only a few refinement steps, but its shape becomes progressively clearer as more refinement iterations are applied. However, for small objects or regions with almost no initial points, we empirically observe that even multiple refinement iterations often struggle to generate sufficiently or recover the missing points. We will add this in the final draft.
>
> ---
>
> ### [W5-2] Could the authors provide results or discussion on the model’s performance in static or near-static scenes, where inter-frame displacements are minimal?
>
> ---
>
> We would like to emphasize that strictly static or near-static scenes with minimal inter-frame displacements are not our primary focus, and existing methods can already be highly effective and more efficient. Nevertheless, as shown by the Neu3DV results (Table W1), SPIN‑4DGS remains competitive: it achieves PSNR comparable to explicit 4DGS and consistently higher than D3DGS. This aligns with our claim: the advantage of SPIN‑4DGS is most pronounced under large inter‑frame displacements, but the method remains effective in low‑motion regimes as well.
>
> ---

---

> > ### Comment · Reviewer_B2ix · 2025-11-27
> >
> > Thank you for the additional quantitative evaluations on Neu3DV and for including the MoDec-GS and LPIPS results. These additions address several of my earlier concerns regarding the breadth and depth of the evaluation. I also appreciate the authors’ clarification of the method’s limitations and the discussion on performance in static or low-motion scenarios.
> >
> > That said, there remain several aspects that need further attention. In particular, the comparison to MoDec-GS is conducted only on the Basketball scene in the Panoptic dataset. It is unclear how the proposed model performs relative to existing state-of-the-art baselines on the remaining sequences, which is important for a fair and comprehensive assessment. In addition, no new qualitative comparisons are provided, making it difficult to fully verify the claimed advantages.
> >
> > Finally, as Reviewer H2vU noted, since the conference allows revised manuscripts during the rebuttal period, it is essential that the newly reported results, expanded discussions, and clarifications be integrated into the paper itself for completeness. For a proper final assessment, I would like to review a revised version of the submission.

---

> ### Author Response · Authors · 2025-11-28
>
> Dear reviewer B2ix,
>
> We appreciate your efforts in reviewing our paper and actively participating in the discussion.
>
> ---
> We would like to clarify that most issues you raised were already addressed in the revised manuscript and included in the PDF that we uploaded a week ago (on 22 Nov). It seems that some of the changes may not have been easily visible, possibly because the color highlighting was not applied in that version. To avoid any confusion for you and for other reviewers/AC, we have now re-uploaded the revised PDF with highlighted in blue.
>
> This highlighted version clearly reflects the revisions made in response to your comments: specifically, we have added the MoDec-GS quantitative results to the main comparison Table 1. As shown in these results, our method consistently achieves higher PSNR than MoDec-GS across all six sports scenes. Furthermore, we have included new qualitative visualizations in Figure 7 in the appendix that illustrate the performance improvements of our method over existing baselines.
>
>
>
> | Method     | Basketball | Boxes | Football | Juggle | Softball | Tennis | AVG   |
> |------------|------------|-------|----------|--------|----------|--------|--------|
> | 4DGaussian | 27.28      | 27.32 | 28.71    | 26.94  | 27.24    | 27.66  | 27.53 |
> | MoDec      | 27.42      | 26.17 | 27.09    | 28.03  | 27.31    | 27.36  | 27.23 |
> | **Ours**   | **30.05**  | **29.91** | **29.99** | **30.31** | **30.24** | **30.14** | **30.11** |
>
> ---
> We apologize for any inconvenience the previous version may have caused and appreciate your careful attention to our submission. Please let us know if any further clarification or additional revisions are needed.
>
> Thank you once again for your thoughtful and constructive review.

---

### Official Review · Reviewer_APBY · 2025-10-29

**Soundness:** 1
**Presentation:** 2
**Contribution:** 2
**Rating:** 2
**Confidence:** 4

**Summary:**

This paper presents SPIN-4DGS,  a practical and well-engineered solution to 4D dynamic spatial reconstruction. This paper observes the defects of previous 4DGS related methods and establishes a method with hybrid explicit and implicit modeling. The proposed method collects the spatial-temporal positions from other 4DGS methods and predicts the other Gaussian attributes based on the positional information in a feed-forward paradigm. This formulation avoids the shared properties across timesteps and achieves higher performance on sports related datasets.

**Strengths:**

1. This paper is well-motived. Presenting highly dynamic areas with large motions has been a long-standing problem in 4D reconstruction. This paper analyzes the origin of this phenomenon and concludes this as the temporally shared features.
2. Satisfying performance. The proposed method maintains high-fidelity reconstruction for large-motion 4D scenes, from both the quantitative results and visual effects.

**Weaknesses:**

1. Method formulation. The proposed pipeline tries to directly capture the spatio-temporal positions from previous 4DGS representations and build a feed-forward network to estimate the other properties. However, the position prior from other methods may introduce bias from individual methods, as different 4DGS methods show different specialties, making this design unstable and unrobust. Besides, this method trains a feed-forward network to estimate the Gaussian attributes to avoid temporally shared properties, which is a proxy of 'per-time 3DGS', i.e., the upper bound of this method is just naively maintaining a set of 3DGS for each timestep (which handles the large-motion areas best), and their method is merely trying to save these per-timestep 3DGS into a larger feed-forward network. In my opinion, this design is not of novelty or significance, and does not figures the core problem out. Besides, this method naively integrates the hash MLP and feed-forward attribute estimations, which is a similar pipeline with Instant Gaussian Stream (CVPR 25), and these techniques have been used in HAC (ECCV 24), 3DGStream (CVPR 24) and feed-forward reconstruction pipelines such as MVGaussian (ECCV 24), TranSplat (AAAI 25).
2. Possible overclaim. This work claims in Line 104 to 106: 'our design preserves the rendering efficiency of prior 4DGS methods while improving storage and training stability, thereby enhancing the efficiency–quality balance required for practical deployment', which introduces confusions: Do we need to inference the feed-forward network multiple times before obtaining the final 4D result, or do we need to inference and store the estimated per-timestep Gaussian attributes in ahead? If the former applies, the method requires network inference before each rendering, thus the rendering speed is highly lagged. If the later applies, the method requires saving per-timestep Gaussian attributes, thus the storage overhead is huge. This leads to an overclaim of the method performance.
2. Training dataset and generality. The proposed evaluation is implemented on the Panoptic dataset, while how the feed-forward network is trained is unclear. Is the training implemented on the same dataset? It is not clear whether this paradigm guarantees a correct evaluation. If the network is trained across all scenes in Panoptic dataset, will it obtain a better performance by per-scene per-training? Besides, it is wondered how this feed-forward network obtains generality as the training and evaluation datasets are of small size. It is also interesting to investigate its performance on other 4D reconstruction benchmark.

**Questions:**

The main concerns are elaborated in the weakness, with confusions on the method formulation, performance claim and the training protocols.

If the authors' rebuttal solves misunderstandings in the review, I am willing to further adjust my score.

---

> ### Author Response · Authors · 2025-11-20
>
> Dear reviewer APBY,
>
> We thank the reviewer for the thoughtful and constructive review, and also appreciate your remark that you are willing to adjust your score if the current misunderstandings are resolved. We will carefully revise the manuscript to clarify these points and improve the overall presentation in line with your suggestions.
>
> ---
>
> The key point we would like to clarify is how SPIN‑4DGS handles temporal sharing. Our method does not avoid temporally shared properties; it reparameterizes them. We first estimate explicit spatiotemporal positions $(x,y,z,t)$ and then use a single shared implicit network $f_{\theta}(x,y,z,t)$ to predict Gaussian attributes for all timesteps. Time ($t \in [0,1]$) is an input to this function, and the parameters $\theta$ are optimized jointly over all frames. Thus, attributes at different times are outputs of the same 4D spatio‑temporal field, rather than independent per‑frame parameters. In contrast, standard explicit 4DGS optimizes a separate attribute vector at each $(x,y,z,t)$, which actually imposes weaker temporal sharing than our field‑based design.
>
> Some phrasing in the current draft may have suggested that our method is merely a well‑engineered 4D dynamic spatial reconstruction or a proxy for per‑time 3DGS. We will revise the paper to make our temporal parameterization explicit and to stress that our contribution is to stabilize temporal sharing under large inter-frame motions via a field-based formulation that operates directly over spatio-temporal samples, without tracking Gaussian identities over time. Below, we respond to W1–W3 in detail and hope these clarifications address the remaining concerns.
>
> ----

---

> ### Author Response · Authors · 2025-11-20
>
> ### [W1-1] Robustness to stage-1 position bias and fast-motion missing
>
> ---
>
> We agree that the quality of the Stage‑1 position estimator affects our final result. However, all current 4DGS methods suffer from missing or redundant Gaussians around fast‑moving objects, so any method that uses 4DGS positions as input necessarily inherits this limitation.
>
>  To mitigate this dependency, we introduce a lightweight refinement stage that performs a few iterations of Gaussian-splatting optimization with densify/prune. This stage removes redundant points and allocates new Gaussians to regions where the Stage‑1 positions are insufficient, especially around fast motions. As shown in Fig. 6, increasing the number of refinement iterations progressively recovers the shape of the tennis racket, which is only partially reconstructed without refinement.
>
>  Similar to 3DGS, regions with absolutely no initial points cannot be fully recovered, but in practice the refinement significantly reduces the bias from the initial estimator and yields clearer reconstructions of fast-moving objects. After refinement, during training of the attribute field we allow small gradient‑based updates to the spatiotemporal positions (with capped step sizes), which further corrects initialization bias around fast motions. Overall, these designs make SPIN‑4DGS much less sensitive to the particular choice of Stage‑1 method and improves robustness, even when the initial position prior is biased or incomplete.
>
> ----
>
> ### [W1-2]: Distinguishing SPIN-4DGS from per-time 3DGS
> ---
>
> We would like to clarify that SPIN‑4DGS does not avoid temporally shared properties, nor is it simply a proxy for per‑time 3DGS. Our model learns a single implicit spatio‑temporal field $f_{\theta}(x,y,z,t)$ that outputs Gaussian attributes for all sampled spatiotemporal positions. The parameters $\theta$ are shared across time, so attributes at different frames are not independent; instead, the network must encode how Gaussian color, opacity, and shape vary across the 4D space–time volume. In contrast, a per‑time 3DGS model optimizes an independent attribute vector for each Gaussian at each timestep, with no parameter sharing across frames.
>
> Importantly, SPIN‑4DGS is not trained by first fitting frame-wise 3DGS and then distilling them into a feed‑forward network. We only use a prior 4DGS method as a position estimator to obtain spatiotemporal Gaussian positions; all Gaussian attributes and the network parameters $\theta$ are initialized from scratch and optimized directly with the rasterization loss. There is no stage where frame-wise 3DGS attributes are used as supervision, so our method is not bounded by a set of 3DGS for each timestep.
>
> Empirically, SPIN‑4DGS consistently outperforms both (per‑time) 3DGS and D3DGS on all six CMU Panoptic Sports scenes, with an average PSNR gain of about 1.4–1.9 dB and up to +3.03 dB on Basketball. If our method were merely maintaining or compressing a set of independent per‑frame 3DGS results, it would be difficult to obtain such consistent and significant improvements.
>
>
> | Model | Basketball | Boxes | Football | Juggle | Softball | Tennis | Mean  |
> |-------|------------|-------|----------|--------|----------|--------|-------|
> | 3DGS  | 27.02      | 28.74 | 28.49    | 28.19  | 28.77    | 28.03  | 28.21 |
> | D3DGS | 28.22      | 29.46 | 28.49    | 29.48  | 28.43    | 28.11  | 28.7  |
> | Ours  | 30.05      | 29.91 | 29.99    | 30.31  | 30.24    | 30.14  | 30.11 |
>
> ---

---

> ### Author Response · Authors · 2025-11-20
>
> ### [W1-3] Core problem and novelty
>
> ---
>
> We appreciate the opportunity to clarify the core problem we address. Our goal is to prevent temporal collapse of Gaussian attributes in dynamic scenes with large inter-frame displacements, such as sports sequences where balls and rackets move rapidly and frequently overlap. In many existing 4DGS methods, Gaussian attributes are shared (or tightly coupled) across time, so under large motions gradients from one frame can interfere with others, making fast-moving objects blurry or even disappear.
>
> SPIN‑4DGS tackles this problem by re-factorizing the 4D representation: we keep explicit spatio-temporal positions and learn a feed-forward network $f_{\theta}(x,y,z,t)$ that predicts per-frame Gaussian attributes. This 4D attribute field can vary flexibly over time while still sharing parameters across frames, and in practice our method maintains stability for fast motions and yields sharper reconstructions than both explicit 4DGS and deformable baselines, especially on challenging sports scenes.
>
> Conceptually, this differs from canonical-space deformable and standard explicit 4DGS, which encode appearance as parameters that are shared (or tightly coupled) across timesteps. In those methods, a single set of Gaussian attributes influences multiple frames simultaneously, so updates driven by one frame can easily degrade others under large motions. Our temporal parameterization instead breaks this direct coupling at the attribute level: different $(x,y,z,t)$ samples can receive different attributes while still being linked through the shared parameters θ. This reduces the temporal collapse diagnosed in our analysis and, in our experiments, keeps SPIN‑4DGS stable under large inter-frame displacements. We keep positions explicit and only allow light per‑sample adjustments during training; the temporal representation itself resides in $f_{\theta}(x,y,z,t)$, not in a point‑to‑point deformation field.
>
> Moreover, several deformable 4DGS methods explicitly avoid time-varying color and opacity, noting that allowing these attributes to change over time tends to produce unstable or implausible surfaces when rendering novel views, and thus makes tracking difficult. In SPIN‑4DGS, by predicting color and opacity from spatiotemporal coordinates via $f_{\theta}(x,y,z,t)$, we can model time-varying appearance while keeping the underlying Gaussian support stable, which is crucial for fast-moving objects with large inter-frame displacements.
>
> ----
>
> ### [W1-4] Related work discussion
>
> ---
>
>  While our method also uses a hash encoder and MLPs, their role and objective are different from the cited works. SPIN‑4DGS uses a shared hash encoder on $(x,y,z,t)$ followed by several lightweight decoders that together produce the full Gaussian parameter set (scale, rotation, opacity, and color) at each spatiotemporal position. These decoders are trained under a schedule specifically designed to keep Gaussians stable around fast motions.
>
>  In contrast, Instant Gaussian Stream relies on additional supervision such as optical flow and does not directly decode full Gaussian parameters from spatiotemporal coordinates $(x,y,z,t)$. HAC, MoDec‑GS and 3DGStream either use anchors or canonical space transformation caches to deform an initial 3D Gaussian set, but they do not generate complete per‑frame Gaussian parameters solely as a function of $(x,y,z,t)$.
>  TranSplat and MVGaussian predict per‑pixel Gaussians for view synthesis, not per‑Gaussian parameters in 4D space.
>
>  Thus, although we share some architectural components (hash grids and MLPs), SPIN‑4DGS adopts a different decoding formulation: it treats $(x,y,z,t)$ as the primary input and learns a spatiotemporal Gaussian parameter field tailored to large‑motion 4D reconstruction without external supervision. In particular, our designs are chosen to keep color and opacity predictions stable over time, addressing the instability that has been reported when deforming these attributes directly in prior deformable 4DGS methods.
>
> ---

---

> > ### Author Response · Authors · 2025-11-20
> >
> > ### [W2] Resolving the Efficiency–Storage Ambiguity in Inference
> >
> > ---
> >
> >
> > We thank the reviewer for pointing out the ambiguity in Lines 104–106. There are indeed two different ways to use our model at inference time, which were not clearly distinguished in the current draft.
> >
> >  (i) Online mode (reported in Table 1): Gaussian attributes are predicted by the feed-forward network on-the-fly, and the reported 104 FPS includes this network inference cost. This setting has minimal storage overhead, since we do not store per-timestep attributes.
> >
> >  (ii) Pre-baked mode: Alternatively, we can run the network once to generate explicit Gaussian attributes for all timesteps and then discard the network. Rendering then becomes pure rasterization, without any further network inference. In this mode, SPIN‑4DGS reaches around 720 FPS, which is roughly 3.5× the throughput of explicit 4DGS under the same rasterizer.
> >
> > Regarding the sentence in Lines 104–106, our intention was not to claim that SPIN‑4DGS is simultaneously as fast as the most lightweight 4DGS configuration and as memory‑efficient as a purely implicit model under all settings. Rather, we aim to offer a favorable efficiency–quality trade-off among methods that can handle large inter-frame motions. In this regime, the most relevant baseline is D3DGS, which is the only prior method that produces similarly sharp reconstructions on our sports scenes. Compared to D3DGS, SPIN‑4DGS is about 1.6× more storage‑efficient, achieves higher reconstruction quality on all six CMU Panoptic Sports sequences, and does not require external supervision, thereby avoiding the additional training cost. We will soften the wording accordingly and clarify that our claim is about this efficiency–quality balance in the large‑motion setting, not about being the single best method in terms of raw speed or storage in isolation.
> >
> > ----

---

> ### Author Response · Authors · 2025-11-20
>
> ### [W3-1] Per-Scene training protocol
>
> ---
>
> We would like to clarify that SPIN‑4DGS is trained per scene, following exactly the same protocol as prior dynamic 4DGS work: for each Panoptic Sports sequence we train a separate model on that sequence and evaluate it on held-out views from the same scene.
> We do not train a single feed-forward network jointly across multiple scenes. Hence the evaluation is standard and directly comparable to existing baselines.
> Based on your suggestion, we also conducted a cross-scene experiment: a model pre-trained on the Football scene and then fine-tuned on the Basketball scene converged faster initially but achieved slightly worse final PSNR than a Basketball model trained from scratch. Exploring larger multi-scene training regimes is an interesting direction for future work but is orthogonal to the main contribution of this paper.
>
> ---
>
> ### [W3-2] Generalization and additional benchmarks
>
> ---
>
> Regarding generality under limited data, we conducted an additional experiment where SPIN‑4DGS was trained using only 15 consecutive frames from the Basketball scene. At test time, we evaluated on test views corresponding to the same time-steps. The performance gap between training with all frames and with only 15 frames was modest (e.g., PSNR decreased by only 0.53 dB). This indicates that the feed-forward network can maintain strong reconstruction quality even when the number of training frames is heavily reduced.
>
> We also evaluated SPIN‑4DGS on another 4D reconstruction benchmark to examine its behavior beyond large-motion sports scenes. While our primary application is fast-moving objects with large inter-frame displacements (as in CMU Panoptic Sports), we additionally ran experiments on the Neu3DV benchmark, which contains scenes with complex backgrounds and relatively slow motions. Across these scenes, SPIN‑4DGS achieves a mean PSNR of 32.19 dB, improving over explicit 4DGS (32.00 dB) by 0.19 dB and over D3DGS (31.04 dB) by 1.15 dB. These results suggest that our method can also achieve fine reconstruction quality in different regimes with slow motions and complex backgrounds,  while D3DGS (the strongest baseline on Panoptic Sports) performs worse than the explicit 4DGS baseline. We will include the Neu3DV results in the revised manuscript. We will include these results in the final manuscript.
>
> | Model | 4DGS Category        | Mean PSNR |
> |-------|----------------------|-----------|
> | D3DGS | External supervision | 31.04     |
> | 4DGS  | Explicit             | 32.00     |
> | SPIN-4DGS (Ours)  | -                    | 32.19     |
>
>
> ---

---

> > ### Comment · Reviewer_APBY · 2025-11-26
> >
> > I appreciate the authors' efforts in preparing this rebuttal, and I have carefully re-examined the rebuttal along with the paper. However, I still have the following comments and remaining concerns:
> >
> > 1. As discussed above, insufficiently clear method formulation and unsuitable performance claim exist in the paper. A thorough reorganization of the entire manuscript may be necessary to improve clarity and ensure appropriate justification.
> >
> > 2. The motivation analysis is not convincing. The main paper concludes that existing 4DGS approaches degrade in fast-motion scenarios due to temporal dependency issues—i.e., "optimizing for one frame makes other slices suboptimal". I agree with this motivation. Nevertheless, it is still unclear how the proposed design effectively addresses this challenge. The rebuttal states that shared temporal features (via the per-scene network) contribute to resolving this issue.
> >
> > 3. Comparison with per-frame 3DGS. In my opinion, per-frame 3DGS can be considered as an upper-bound for 4D reconstruction. For example, optimizing each timestep can totally mitigate the problem of blurry representation for fast-moving objects. It is implausible and unconvincing that the proposed method can yield 3 dB higher fidelity over per-frame 3DGS without further justification. It is hard to understand how the proposed method can yield such good performance.
> >
> > 4. On the initial spatial-temporal positions. Does this imply that a separate 4DGS model must be trained beforehand for obtaining the spatial-temporal positions, before the proposed method is trained?
> >
> > 5. Novelty relative to prior work. As discussed, the use of a hash network to fit Gaussian attributes does not appear substantially different from prior methods. The contribution would benefit from a clearer clarification of what is technically novel and what advantages are uniquely introduced by such proposed approach.

---

> ### Author Response · Authors · 2025-11-28
>
> Dear reviewer APBY,
>
> We appreciate your efforts in reviewing our paper and actively participating in the discussion.
>
> ---
>
> ###  Q1 : Method clarity and performance claim justification
> We appreciate the reviewer’s feedback that the original version did not present the method formulation and performance claims with sufficient clarity. In the revised manuscript, we have clarified and reorganized the key parts of the Introduction, Method, and Experiments sections, and added further explanations in the Appendix.
> All modified parts are highlighted in blue in the revised manuscript. We hope that below clarifications and updated revisions address your concerns about clarity and justification.
>
> ----
>
> ###  Q2 :  Unclear explanation of how shared temporal features address the challenge
>
> We appreciate the reviewer’s concern and would like to provide more explanations. The key point we would like to clarify is where temporal sharing happens. Existing 4DGS methods share explicit Gaussian attributes (color, opacity, etc.) across all time slices of the same 4D Gaussian. A single attribute vector must simultaneously explain all frames in which that Gaussian is visible; as a result, optimizing for one frame directly overwrites the attributes used by other slices, and background Gaussians that are visible in many frames dominate the updates.
>
> In contrast, SPIN‑4DGS does not share attributes themselves across time. Instead, we learn a per‑scene implicit function :  $f_{\theta} : (x, y, z, t) \mapsto \mathbf{a}$ that maps spatiotemporal positions to Gaussian attributes. Attributes at different timesteps are no longer tied by a single vector; they are input‑conditioned outputs of the same function. Thus, temporal sharing occurs at the feature‑space of $f_{\theta}$, rather than at the level of a single attribute vector. This is analogous to 4D NeRF, where a single network $f_{\theta}(x, y, z, t)$ jointly fits all timesteps but is still able to represent different colors/densities at different times.
>
> Because of this change in parameterization, gradients from different frames no longer compete over the same explicit attribute vector. Each spatiotemporal position $(x, y, z, t)$ obtains its own attributes through $f_{\theta}$, and optimization updates the shared basis functions so that they jointly explain all timesteps. This converts the “hard tying” of attributes across time into input‑conditioned soft sharing, which empirically prevents the collapse of fast‑moving Gaussians even under large inter‑frame displacements.
>
> To further show that SPIN‑4DGS learns temporal dynamics rather than per‑frame spatial attributes, we conduct a leave‑last‑frame experiment on a 150‑frame video. We train ours and 4DGaussian methods only on the first 149 frames and completely exclude frame 150 from training. At test time, we provide the Gaussian positions for frame 150 to infer the attributes for this unseen frame purely from the learned temporal representation.
> Under this protocol, SPIN‑4DGS achieves higher PSNR on frame 150 than 4DGaussian, even though both methods see exactly the same training frames. This result suggests that our network $f_{\theta}(x, y, z, t)$ has learned a time-conditioned 4D attribute field: even for a frame that is completely unseen during training, it can still infer non-trivial Gaussian attributes from spatiotemporal positions alone and achieves higher PSNR than 4DGaussian (deformable baseline) under the same protocol. Although the absolute PSNR on this withheld frame is moderate, it is already higher than the full-scene PSNR achieved by Grid4D in Table 1 (trained on all 150 frames), indicating that SPIN-4DGS captures temporal dynamics that enable meaningful generalization to unseen timesteps.

---

> ### Author Response · Authors · 2025-11-28
>
> ### Q3 : Insufficient justification for improvements over per-frame 3DGS
>
> ---
>
> We respectfully disagree with the premise that per-frame 3DGS provides an upper bound on 4D reconstruction quality. Per-frame 3DGS can indeed remove temporal coupling and thus avoid blur by treating each timestep as an independent static scene. However, it has no mechanism to exploit information from other frames. In realistic dynamic scenes with limited cameras, each frame by itself may provide insufficient information: important surfaces may be occluded or poorly conditioned, and a per-frame model can easily memorize its own training views while producing geometry and appearance that do not transfer well to novel viewpoints.
>
> To demonstrate this, we additionally conducted a new experiment: for a given frame t₀, we trained the explicit baselines 3DGS only on that single frame, following the standard per-frame setting, so that all of their capacity was devoted to fitting this timestep. In parallel, we also train a single-frame variant of SPIN-4DGS (“Ours, Single-frame”) on the same frame t₀. Finally, we train our full method (“Ours, Full frames”) on the entire sequence, which must share its parameters across all frames. We then evaluate all methods on held-out (novel) views of the same frame t₀. As shown in the table below, 3DGS achieves very high PSNR on the training views of t₀, but its PSNR on the novel views is significantly lower than that of SPIN-4DGS trained on the full sequence (“Ours, Full frames”), since SPIN-4DGS has learned all frames jointly. Moreover, the single-frame variant of our method (“Ours, Single-frame”) attains the highest PSNR on the training views but the lowest PSNR on the novel views among all variants, indicating that it also strongly overfits when temporal information is removed. This supports that per-frame 3DGS (or our variant) behave likely to memorize training views, not provide satisfying performances on novel views. Therefore, we believe that ours clearly leverages temporal information and thus achieves high-quality 4D reconstruction.
>
> Moreover, consistent with our observations, prior work such as “Representing Long Volumetric Video with Temporal Gaussian Hierarchy” [1] already reports that 4D Gaussian representations that exploit temporal structure can outperform per-frame 3DGS on benchmarks like Neu3DV in terms of reconstruction quality. This empirical trend suggests that 4D methods can better leverage temporal redundancy and multi-view constraints across frames to obtain more consistent geometry and appearance, whereas per-frame 3DGS optimizes each timestep in isolation and is therefore more susceptible to overfitting to the particular training cameras.
>
> Consequently, the PSNR improvements we observe over per-frame 3DGS (approximately +2 dB on average across all scenes and up to +3 dB on the basketball scene) are fully consistent with both our additional experiments and prior literature. In many cases, SPIN-4DGS even attains slightly lower training performances compared to baselines (e.g., 4DGS) while clearly outperforming them on the test views, which is precisely the behavior expected from a model that generalizes better. These gains arise from learning shared spatiotemporal features across multiple frames and viewpoints, which mitigates per-frame overfitting and improves novel-view performance through a coherent 4D representation.
>
> | Method | Setting      | Train | Test  |
> |--------|--------------|-------|-------|
> | 3DGS   | Single-frame | 42.26 | 28.38 |
> | Ours   | Single-frame  | 43.48 | 27.23 |
> | Ours   | Full frames  | 37.80 | 29.72 |
>
>
> ----
>
> [1] Representing Long Volumetric Video with Temporal Gaussian Hierarchy’ SIGGRAPH Asia 2024

---

> ### Author Response · Authors · 2025-11-28
>
> ### Q4 : Requirement of a separately trained 4DGS model for obtaining initial spatio-temporal positions
> ---
>
> We do not require a separately trained 4DGS model. In our pipeline, 4DGS is used only as a lightweight initializer for the spatio-temporal positions: we run 4DGS for 15K iterations (about 10 minutes), stop optimization there, and keep only the Gaussian positions as initialization for Stage-2. No further training or rendering with 4DGS is performed.
>
> The table below already includes this warm-start cost in the reported training times, and shows that the overall pipeline is competitive or even more efficient than standard 4D baselines. In the “15K, no refine” setting, SPIN-4DGS reaches 28.82 dB in 45 minutes (10 min for position + 35 min for the attribute network), which is both faster and more accurate than 4DGS (27.89 dB in 60 minutes) and D3DGS (28.22 dB in 100 minutes). If one is willing to spend slightly more time, the “15K, refine (0.5K)” variant achieves 29.13 dB in 73 minutes, still with shorter total training time than D3DGS while providing clearly higher fidelity. Higher-budget variants such as “15K, refine (2K)” and “40K, refine (2K)” further improve PSNR at the cost of additional compute.
>
> Therefore, SPIN-4DGS does not rely on training a full 4DGS model as a separate pre-processing step. Instead, we use a brief 4DGS warm-up to estimate initial spatio-temporal positions and then learn all Gaussian attributes with a feed-forward network. This design keeps the additional overhead minimal while still providing a favorable trade-off between reconstruction quality and training time across different compute budgets.
>
> | Method    | Setting             | PSNR  | Position (min) | Refine (min) | Network (min) | Total (min) |
> |-----------|---------------------|-------|----------------|--------------|---------------|------------:|
> | 4DGS      | default             | 27.89 | –              | –            | –             |         60  |
> | D3DGS     | default             | 28.22 | –              | –            | –             |        100  |
> | SPIN-4DGS | 15K, no refine      | 28.82 | 10             | 0            | 35            |         45  |
> | SPIN-4DGS | 15K, refine (0.5K)  | 29.13 | 10             | 33           | 30            |         73  |
> | SPIN-4DGS | 15K, refine (2K)    | 29.51 | 10             | 100          | 32            |        142  |
> | SPIN-4DGS | 40K, refine (2K)    | 30.05 | 10             | 100          | 85            |        195  |

---

> ### Author Response · Authors · 2025-11-28
>
> ### Q5 : Technical novelty and uniquely introduced advantages
>
> ---
>
> We believe that the main technical contribution of SPIN-4DGS lies in how the implicit network is used to reparameterize and train 4D Gaussian attributes, rather than in the choice of network backbone itself. Concretely, after a short 4DGS warm-up to estimate spatio-temporal positions (Stage 1), we discard all appearance parameters from the warm-up and treat color, opacity, scale, and rotation as outputs of a single 4D implicit field conditioned only on $(x, y, z, t)$ (Stage 2). The rest of our design, particularly the attribute-aware decoder, is tailored to make this from-scratch attribute learning stable and effective.
>
> This stage-2 formulation is fundamentally different from prior 4DGS pipelines that are built around pre-defined Gaussian attributes. Existing 4DGS approaches typically assume that a good set of per-Gaussian attributes has already been obtained in a static setting (e.g., from a canonical reconstruction or from a first-frame model initialized by SfM). These pre-optimized colors, opacities, scales, and rotations are then propagated or slightly adjusted over time, so that temporal modeling mainly amounts to warping positions rather than learning the full 4D attribute field. In SPIN-4DGS, we still use a brief 4DGS warm-up to populate the 4D domain with Gaussian centers, but we explicitly throw away all appearance information from this stage and force the implicit network to reconstruct every attribute purely from spatio-temporal positions. This position-only conditioning removes any reliance on canonical or first-frame appearance estimates, so SPIN-4DGS is not bottlenecked by errors or missing attributes from those unreliable estimations.
>
> Learning all Gaussian parameters from scratch in this way is also a much harder optimization problem than refining pre-defined attributes. Scale, opacity, rotation, and color live on very different numerical ranges and manifolds (e.g., positive scales, bounded opacities, rotations, colors from spherical harmonics), and a simple configuration for those attribute decoders typically leads to ill-conditioned gradients, training instability, or severe artifacts. Our attribute-aware decoder mitigates this issue by using separate branches with tailored parameterizations for different attribute groups, along with activations and constraints that respect the numerical range and manifold of each attribute. This design is particularly substantial for color and opacity: prior deformable 4DGS methods often keep these attributes fixed or nearly time-invariant for stability, whereas SPIN-4DGS predicts fully time-varying color and opacity as outputs of $f_{\theta}(x, y, z, t)$ and remains stable in practice.
>
> In summary, the novelty of SPIN-4DGS relative to prior 4DGS work is twofold: (i) we reparameterize 4D Gaussian attributes as a position-conditioned implicit field learned from scratch, instead of relying on canonical or first-frame Gaussian attributes as the primary carrier of appearance, and (ii) we introduce an attribute-aware decoder that makes this full attribute prediction problem numerically stable, enabling high-fidelity, time-varying appearance even under fast, large-displacement motion.

---

### Official Review · Reviewer_9M6m · 2025-10-31

**Soundness:** 3
**Presentation:** 2
**Contribution:** 3
**Rating:** 6
**Confidence:** 4

**Summary:**

This paper addresses the failure of 4D Gaussian Splatting (4DGS) to reconstruct scenes with fast motion and large inter-frame displacements, where objects often blur or disappear. The authors propose SPIN-4DGS, a novel framework that decouples spatiotemporal position estimation from Gaussian attribute learning. The core contribution is a hybrid approach where explicit 4D spatiotemporal positions (x, y, z, t) are used as inputs to a lightweight implicit network (a 4D hash encoder and MLP decoders) that predicts all other Gaussian attributes (e.g., scale, rotation, color, opacity). This design avoids the attribute collapse seen in prior work and demonstrates state-of-the-art reconstruction fidelity on challenging dynamic sports datasets.

**Strengths:**

* problem definition and diagnosis: The paper correctly identifies a significant and open challenge for the 4DGS paradigm: robustly modeling high-frequency dynamics. The analysis of failure modes in prior art—distinguishing between deformable methods (canonical space initialization failure ) and explicit 4D parameterizations (attribute collapse due to cross-frame interference )—is insightful and provides a methodologically sound basis for the proposed solution.
*  novelty of the implicit attribute network: The core idea of decoupling explicit positions from implicitly-learned attributes is novel. This hybrid representation allows the model to leverage the efficiency of explicit splatting while using the generalization power of an implicit network to maintain spatio-temporal consistency for attributes. This directly addresses the attribute collapse observed in prior explicit 4DGS methods .
* State-of-the-Art reconstruction fidelity: The method achieves significant quantitative and qualitative improvements on the CMU Panoptic Sports dataset, a highly appropriate and challenging benchmark for this task. The gains over strong baselines, including those using external supervision (e.g., +1.83 dB PSNR over D3DGS on 'Basketball'), are substantial and clearly demonstrate the method's effectiveness in capturing fast-moving objects.

**Weaknesses:**

* Efficiency-Quality Trade-off: The claims regarding rendering efficiency  are not fully substantiated and appear to be a key limitation. Table 1 indicates a significant drop in rendering speed (104 FPS) compared to the explicit 4DGS baseline (197 FPS) and especially deformable methods. This is likely due to the computational overhead of querying the implicit network for all Gaussian attributes per-frame during rasterization, which is a non-trivial trade-off for the gains in fidelity. This trade-off should be discussed more explicitly.
* Storage Footprint: The paper's claim of "reducing storage overhead"  appears overstated. The reported storage (1261MB) is only marginally better than the 4DGS baseline (1293MB). This suggests the bulk of the memory is consumed by the explicit spatiotemporal positions, and the implicit network for attributes does not offer a significant storage advantage in practice. The contribution lies in enabling reconstruction, not in compressing the representation.
* Training Scalability and Cost: The method introduces a two-stage training process. The total training cost (Stage 1 position estimation  + Stage 2 refinement  + Stage 2 implicit network training) relative to end-to-end baselines is not clearly reported. This makes it difficult to assess the practical scalability and computational budget required to deploy this method.

**Questions:**

* Q1:Inference Cost Analysis: Could the authors confirm that the ~2x reduction in FPS (vs. 4DGS) is due to the inference cost of the 4D Hash Encoder and MLP decoders required for every Gaussian at render time? If so, this is a critical aspect of the method's efficiency profile and warrants clarification.
* Q2: Storage Claims: Given the data in Table 1, the storage benefit appears negligible. Would the authors consider reframing this contribution? The novelty appears to be in the hybrid representation enabling high-fidelity reconstruction, not in achieving superior compression.
* Q3: Total Training Budget: Please provide a comparative analysis of the total wall-clock training time for SPIN-4DGS (including all stages) versus the primary baselines (e.g., 4DGS, D3DGS) for a single scene.
* Q4: Robustness of Stage 1: The method's success seems dependent on the Stage 1 position estimator. How does SPIN-4DGS handle cases where the explicit estimator (4DGS) completely fails to initialize any points for a fast-moving object? Does the refinement step (Sec 2.2)  only densify/prune, or can it also create new Gaussians in unobserved regions?

---

> ### Author Response · Authors · 2025-11-20
>
> Dear reviewer 9M6m,
>
> Thank you for the clear summary and for highlighting the problem diagnosis, the novelty of decoupling explicit positions from an implicit attribute field, and our state‑of‑the‑art results on fast‑motion sports scenes. We will address your concerns and questions in the response below.
>
> ---
>
> ### [W1 & Q1] Efficiency–quality trade-off
>
> ---
>
>  We thank the reviewer for raising this point. Contrary to the reviewer’s concern, we would like to clarify that the network inference cost can be incurred only once to produce the explicit Gaussian parameters. After this initial feed-forward step, rendering would be performed solely by rasterizing the predicted Gaussians without any further network inference. In this setting, our method reaches around 720 FPS, i.e., nearly 3.5× higher rendering throughput than explicit 4DGS. We note that the number reported in Table 1 (104 FPS) corresponds to the stage where Gaussian attributes are first inferred by the network. We will revise the paper for clarification. As shown in Fig. 4, explicit 4DGS tends to miss fast-moving objects, so we argue that a direct efficiency comparison with explicit 4DGS alone is not entirely fair. When compared with D3DGS, the strongest baseline capable of handling fast motion, our method improves storage-efficiency by at least 1.58×, delivers superior reconstruction quality, and does not require segmentation supervision, thereby avoiding the associated extra training cost. We believe this better reflects the efficiency–quality profile of our method, and we will also revise in the final draft.
>
> ----
>
> ### [W2 & Q2] Storage footprint and claims
>
> ---
>
> Thank you for acknowledging our contribution that enables high-fidelity reconstruction.
>
> Storage: We would like to emphasize again that assessing storage efficiency solely through a comparison with explicit 4DGS may not be entirely fair. The two methods differ substantially in their ability to handle fast motions, which directly affects the real applications in practice. When compared instead with D3DGS, which is relatively capable of capturing fast motion, our method is at least 1.58× more storage-efficient, achieves higher reconstruction quality, and does not require segmentation supervision, thereby avoiding the additional training cost associated with that method. This comparison more accurately reflects the storage–quality characteristics of our method.
>
> Memory: Although our design contains 4D spatiotemporal positions, SPIN-4DGS is trained and rendered in a frame-wise manner: at each optimization step we only rasterize and update Gaussians belonging to a single frame. We never need to keep all per-frame positions in GPU memory at once. As a result, the peak memory usage per step scales with the number of Gaussians in one frame, rather than with the total number of Gaussians across all timesteps. We will revise our claim in the final draft to address these concerns.

---

> ### Author Response · Authors · 2025-11-20
>
> ### [W3 & Q3] Training scalability and total training budget
>
> ---
>
> For practical deployment scenarios with tighter computational budgets, we emphasize that the refinement stage can be substantially shortened (or even omitted) while still yielding competitive performance, since Stage 1 already produces a high-quality explicit initialization and Stage 2 mainly serves to further boost fidelity as shown in Figure 6 of the draft. In this sense, SPIN-4DGS offers a flexible training budget, making it suitable both for high-fidelity offline reconstruction and for the real deployment.
>
> | Method    | Setting             | PSNR  | Position (min) | Refine (min) | Network (min) | Total time       |
> |-----------|---------------------|-------|----------------|--------------|---------------|------------------|
> | D3DGS     | Default             | 28.22 | –              | –            | –             | 1h 40m (100 min) |
> | SPIN-4DGS | 15K, No refine      | 28.82 | 10             | 0            | 35            | 45m              |
> |           | 15K, Refine (0.5K)  | 29.13 | 10             | 33           | 30            | 1h 13m (73 min)  |
> |           | 15K, Refine (2K)    | 29.51 | 10             | 100          | 32            | 2h 22m (142 min) |
> |           | 40K, Refine (2K)    | 30.05 | 10             | 100          | 85            | 3h 15m (195 min) |
>
>
> ----
>
> ### [Q4] Robustness of stage 1
>
> ---
>
> We thank the reviewer for this insightful question about the robustness of Stage 1. While fully recovering regions that have no initial points at all remains intrinsically challenging, our design explicitly aims to progressively mitigate the dependence on the initial position estimator through a dedicated refinement stage. Concretely, the refinement stage performs several iterations of rasterization-based optimization starting from the initialized positions, during which Gaussian densification (densify/prune) gradually removes unnecessary points and allocates new Gaussians in regions that require more detail. In other words, the refinement step does not only prune, but also creates new Gaussians around existing ones in areas with high residuals, thereby incrementally filling in missing structure whenever there is at least some initial support.
>
> For example, Figure 6 in the draft illustrates the effect of the number of refinement iterations on the reconstruction quality in the tennis scene: the racket is only partially reconstructed with a few refinement steps, whereas its shape becomes progressively clearer as more refinement iterations are applied. This demonstrates that our refinement procedure can substantially reduce the dependency on the Stage 1 position estimator and enables clearer reconstruction of fast-moving objects in practice. Meanwhile, in extreme cases where very small and fast-moving objects fail to produce any valid initialization in Stage 1, our method can also miss them; we note that such cases are a common failure mode shared by current Gaussian splatting approaches, including 3DGS/4DGS, which similarly rely on a reasonable point initialization.

---

### Author Response · Authors · 2025-11-20

Dear reviewers and AC,

We sincerely appreciate your time and thoughtful evaluations. Across the reviews, all reviewers noted that our work tackles an important gap in reconstructing scenes with fast motions and large inter-frame displacements (9M6m, APBY, B2ix, H2vU), and that our design is both novel and empirically effective (9M6m, APBY, B2ix). Reviewers also highlighted the clarity of the paper and the thorough ablation studies, as well as state-of-the-art results on the CMU Panoptic Sports dataset (9M6m, B2ix, H2vU).

We appreciate your constructive comments and insights on our manuscript.
We strongly believe that SPIN‑4DGS can be a useful addition to the ICLR community, in particular, due to the enhanced manuscript by reviewers’ comments helping us better deliver the effectiveness of our method.

Thank you very much!
Authors

---

### Author Response · Authors · 2025-12-01

### Evaluation on diverse scenarios

---

Dear reviewers and AC,

We also evaluate SPIN-4DGS on the MeetRoom benchmark as well as Neu3DV, which further addresses Reviewer B2ix’s concern about evaluation on diverse scenarios. The MeetRoom dataset contains cluttered indoor backgrounds and relatively small motions (Discussion, Trimming, VR Headset) scenes. The table below reports PSNR on the MeetRoom scenes: SPIN-4DGS attains a mean PSNR of 32.04 dB, outperforming StreamRF (26.72 dB), 4DGS (30.47 dB), and 3DGStream (30.79 dB) by about +5.3 dB, +1.6 dB, and +1.3 dB, respectively. Qualitative comparisons in Figure 10 further support these findings, showing that while the 4DGS baseline exhibits noise and artifacts in cluttered indoor scenes, SPIN-4DGS yields cleaner and more coherent reconstructions. Evaluating our model across these diverse scenarios, our results show that the SPIN-4DGS remain robust even in scenes with cluttered backgrounds and relatively small motions, which we believe substantially alleviates the concern regarding the diversity of evaluation scenarios. We have revised the manuscript accordingly, and the detailed results are provided in the appendix.

| Method | AVG |
|--------|-----------|
| StreamRF   | 26.72     |
| 4DGS   | 30.47     |
| 3DGStream   | 30.79     |
| Ours   | **32.04** |

---

### Meta-Review · Area_Chair_xVp4 · 2026-01-04

**Summary:**

The paper presents SPIN-4DGS, an explicit 4D Gaussian Splatting framework designed to overcome the limitations of temporally coupled or deformation-based 4DGS methods under fast-motion scenarios. The approach is technically sound and demonstrates empirical improvements, while maintaining a relatively simple and memory-efficient design. The main concerns focus on the limited evaluation scope, the positioning of the contribution with respect to recent explicit 4DGS methods, and the degree of novelty beyond concurrent work. The rebuttal addresses most of these issues by adding new comparisons, metrics, and analyses. Overall, the contribution is solid, and the recommended decision is accept.

**Reviewer Concerns:**

Reviewers generally acknowledge the empirical improvements on fast-motion sequences and the benefits of a simple and memory-efficient design. However, multiple reviewers raised concerns regarding the limited evaluation scope, the clarity of positioning relative to recent explicit 4DGS approaches, and the extent of novelty beyond concurrent work. In response, the rebuttal adds comparisons with MoDec-GS and 4D-Rotor-Gaussians, introduces LPIPS as an additional perceptual metric, and extends the evaluation to Neu3DV. These additions largely address concerns related to general applicability and differentiation.

**Reviewer Scores:**

The paper initially received one reject, one borderline reject, and two borderline accept scores. Based on the rebuttal and subsequent discussion, most reviewer concerns are largely addressed. As a result, reviewers may moderately increase or maintain their original assessments. In particular, Reviewer APBY (who initially rated the paper as reject) is likely to increase their score.

---

### Decision · Program_Chairs · 2026-01-26

Accept (Poster)